# High frequency neural spiking and auditory signaling by ultrafast red-shifted optogenetics

Thomas Mager[1], David Lopez de la Morena[2,3], Verena Senn[4,5], Johannes Schlotte[1,9], Anna D´Errico[1,10], Katrin Feldbauer[1,11], Christian Wrobel[2], Sangyong Jung[2,12], Kai Bodensiek[2], Vladan Rankovic[2,6], Lorcan Browne [2,6,7], Antoine Huet[2,6], Josephine Jüttner[8], Phillip G. Wood[1], Johannes J. Letzkus[4], Tobias Moser [2,3,6] & Ernst Bamberg[1]

Optogenetics revolutionizes basic research in neuroscience and cell biology and bears potential for medical applications. We develop mutants leading to a unifying concept for the construction of various channelrhodopsins with fast closing kinetics. Due to different absorption maxima these channelrhodopsins allow fast neural photoactivation over the whole range of the visible spectrum. We focus our functional analysis on the fast-switching, red light-activated Chrimson variants, because red light has lower light scattering and marginal phototoxicity in tissues. We show paradigmatically for neurons of the cerebral cortex and the auditory nerve that the fast Chrimson mutants enable neural stimulation with firing frequencies of several hundred Hz. They drive spiking at high rates and temporal fidelity with low thresholds for stimulus intensity and duration. Optical cochlear implants restore auditory nerve activity in deaf mice. This demonstrates that the mutants facilitate neuroscience research and future medical applications such as hearing restoration.

[1] Department of Biophysical Chemistry, Max Planck Institute of Biophysics, D-60438 Frankfurt, Germany. [2] Institute for Auditory Neuroscience and InnerEarLab, University Medical Center Göttingen, D-37075 Göttingen, Germany. [3] Göttingen Graduate School for Neuroscience and Molecular Biosciences, University of Göttingen, D-37075 Göttingen, Germany. [4] Neocortical Circuits Lab, Max Planck Institute for Brain Research, D-60438 Frankfurt, Germany. [5] Ernst-Strüngmann-Institute for Neuroscience, D-60528 Frankfurt, Germany. [6] Auditory Neuroscience and Optogenetics Group, German Primate Center, D-37075 Göttingen, Germany. [7] UCL Ear Institute, University College London, London WC1X 8EE, United Kingdom. [8] Friedrich Miescher Institute for Biomedical Research, CH-4058 Basel, Switzerland. [9] Present address: Biozentrum, University of Basel, CH-4056 Basel, Switzerland. [10] Present address: Buchmann Institute of Molecular Life Sciences, Goethe Universität Frankfurt, D-60438 Frankfurt, Germany. [11] Present address: Max-Planck-Institut für Herz- und Lungenforschung, D-61231 Bad Nauheim, Germany. [12] Present address: Neuro Modulation and Neuro Circuitry Group, Singapore Bioimaging Consortium (SBIC), Biomedical Sciences Institutes, A*STAR, 138667 Singapore, Singapore. These authors contributed equally: Thomas Mager, David Lopez de la Morena Correspondence and requests for materials should be addressed to T.M. (email: tmoser@gwdg.de) or to E.B. (email: ernst.bamberg@biophys.mpg.de)

Microbial-type rhodopsins, light gated cation channels (Channelrhodopsins, ChRs) and light-driven ion pumps are useful tools for multimodal optogenetic control of electrically excitable cells in culture, tissue and living animals[1–3]. Since the first description of the ChRs in 2002 and 2003, a set of different ChRs including red-shifted variants like VChR1, ReaChR and Chrimson have been described[4–8]. For different purposes ChRs were modified with respect to the kinetics, ion selectivity as well as light absorption[9–11]. ChR kinetics is a major issue, because the light sensitivity is regulated via the open lifetime of the channel[12]. Channels with a short open lifetime need correspondingly stronger light than channels with a long open lifetime for maximal photostimulation. This is due to the essential invariance of other channel parameters like single channel conductance and quantum efficiency. The mutual dependence between channel kinetics and light sensitivity accounts for the optimization of ChR expression and light delivery for successful experiments in the high frequency range. Although fast channels need stronger light for the activation, high speed is indispensable for many optogenetic applications in neurobiology because many types of neurons operate at high firing rates in the intact animal.

Prominent examples include spiral ganglion neurons (SGNs) of early auditory pathway and fast spiking interneurons in cortical areas, which fire action potentials at up to several hundred Hz[13,14]. However, light stimulation of ChR2-expressing SGNs indicated a strong limitation of the temporal response fidelity[15]. Therefore, fast ChRs are needed and their benefit for use in auditory research has already been indicated using Chronos, a "fast" blue light absorbing ChR, for stimulation of the cochlear nucleus[16].

Electrical cochlea implants (eCI), to date, enable speech understanding in most of approximately 500,000 otherwise deaf users. However, the bottleneck of eCI is the poor frequency resolution of coding that results from wide current spread from each electrode contact and limits speech understanding in background noise[17]. Optical cochlear implants (oCI)-stimulating optogenetically modified SGNs, promise a fundamental advance of prosthetic sound coding by increasing frequency resolution, because light can be better confined than the electric field of electrodes[15]. For oCI eventually to be translated into the clinic, opsins need to be delivered into the SGNs by postnatal virus application to the ear and should endow SGNs with high light-sensitivity and temporal fidelity of spike generation, while light scattering and blue light induced phototoxicity should be minimized. Due to the aforementioned, adverse effects of optogenetic stimulation using blue light, the already available, fast blue light-activated ChR variants like ChETA ($\tau_{off} = 4.4$ ms[9]) and Chronos ($\tau_{off} = 3.6$ ms[6]) might have a limited applicability in animals and future clinical translation.

Here, we report that fast gating can be generally conferred to ChRs by helix 6 (helix F) mutation and demonstrate the utility of fast red-shifted ChRs for driving spiking of fast cerebral interneurons to the limit of their encoding range. Moreover, we established efficient virus-mediated delivery and expression of a fast Chrimson mutant in SGNs of mice, show that single-channel oCIs enable near-physiological spike rates and spike timing in SGNs and restore auditory activity in deaf mice. We demonstrate on several cell types in vitro and in vivo that the unfavorable low light sensitivity for activation is compensated by high expression levels of the fast Chrimson mutants.

## Results

### Fast helix F mutants and their calcium permeabilities. Closed to open state transition is associated with movement of helix F in

several microbial-type rhodopsins[18–20]. Thereby helix F movement controls protonation reactions during vectorial proton transport and consequently the cycle time[21,22]. Closed to open state transitions of helix F have recently been verified for ChR 2 [23,24]. Motivated by these findings we performed a systematic study about the effects of helix F mutations on the closing kinetics of ChR (Fig. 1). We heterologously expressed ChRs helix F mutants in neuroblastoma-glioma cells (NG cells) and performed whole-cell patch-clamp experiments. The helix F mutant F219Y significantly accelerated the closing kinetics of ChRs 2 (Fig. 1c and Table 1). Mutations at the homologous positions of VChR1 (F214Y), ReaChR (F259Y) and Chrimson (Y261F) also accelerated the closing kinetics (Fig. 1b), albeit to a different extent (Fig. 1d–f and Table 1). The strongest effect on the lifetime of the channels was observed in ReaChR and VChR1, where the closing kinetics is accelerated by one order of magnitude (Table 1).

Interestingly, the relative calcium permeability of ChR2 F219Y $P_{Ca}/P_{Na} = 0.30 \pm 0.02$ $(n = 4)$ was increased compared to the relative calcium permeability of ChR2 wt $P_{Ca}/P_{Na} = 0.13 \pm 0.01$ $(n = 4)$. Permeability ratios were calculated according to the Goldman–Hodgkin–Katz equation[25] with the measured values of the reversal potentials after replacing external sodium by calcium. The critical role of a tyrosine at the homologous position on the calcium permeability is verified in ReaChR and Chrimson (Supplementary Table 1). Of note F219 (ChR2 numbering) points to L132 (ChR2 numbering) on helix C in the chimera C1C2 crystal structure (Fig. 1a). ChR2 L132C has an increased calcium permeability (CatCh, calcium translocating ChRs)[10]. In contrast to the FY mutations on helix F, which accelerate the closing kinetics the L132C mutation (helix C) as well as the corresponding mutations at the homologous positions of VChR1, ReaChR and Chrimson significantly slowed the closing kinetics (Table 1). Structural information, the effect on the kinetics and the effect on the calcium permeability indicate a probable interaction of helix C and helix F at those critical residues.

### Chrimson mutants with accelerated closing kinetics. As shown above the Y261F mutation speeds up channel closing in Chrimson (Fig. 1f, Table 1). We identified two additional helix F mutations, which accelerated Chrimson's closing kinetics, namely S267M and Y268F (Fig. 1b, g and Table 1). The combination of the helix F mutations had a cumulative effect, further accelerating channel closing by up to one order of magnitude (Fig. 1g and Table 1). Chrimson mutants carrying the Y268F mutation showed reduced expression in NG cells and a hypsochromic shift of their action spectra by 11 nm (Supplementary Fig. 1 and Supplementary Table 2). The hypsochromic shift might result from an interaction of F268 with the polyene chain of the retinal, as this was shown for F265 (F226, ChR2-numbering), located at the homologous position in the C1C2 structure (Fig. 1a).

Of special interest for optogenetic applications are the fast mutant Chrimson Y261F/S267M (f-Chrimson) and the very fast mutant Chrimson K176R/Y261F/S267M (vf-Chrimson), which carries the additional K176R mutation (Fig. 1h). As described earlier[6], the closing kinetics of Chrimson K176R (ChrimsonR) is accelerated by a factor of ~2 compared to wildtype (Table 1). The closing kinetics of f-Chrimson were strongly accelerated from $\tau_{off} = 24.6 \pm 0.9$ ms (wt-Chrimson, $n = 5$) to $\tau_{off} = 5.7 \pm 0.5$ ms (f-Chrimson, $n = 5$). At the same time f-Chrimson was highly expressed in NG cells (Supplementary Table 2). Vf-Chrimson had ultrafast closing kinetics of $\tau_{off} = 2.7 \pm 0.3$ ms $(n = 7)$, which is at least as fast as the closing kinetics of Chronos

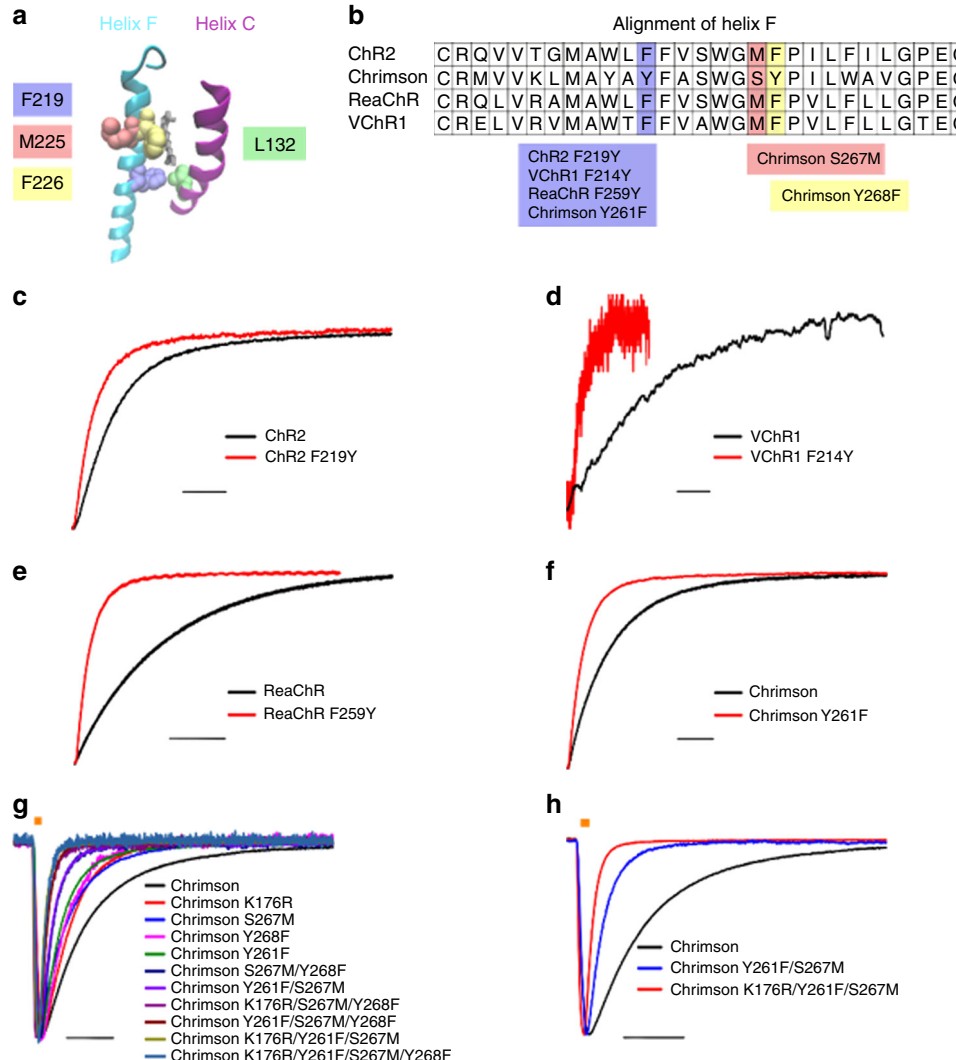

**Fig. 1** Channelrhodopsin mutants with accelerated closing kinetics. **a** Helix F and helix C of channelrhodopsin[48]. Residues changing the off-kinetics are highlighted (ChR2 numbering). **b** ClustalW alignment of the helix F of ChR2, Chrimson, ReaChR and VChR1. Colored boxes show the channelrhodopsin mutants. **c–h** NG cells heterologously expressing channelrhodopsin variants were investigated by whole-cell patch-clamp experiments at a membrane potential of −60 mV. Typical photocurrents of ChR2-EYFP (black trace), ChR2-EYFP F219Y (red trace) (**c**), VChR1-EYFP (black trace), VChR1-EYFP F214Y (red trace) (**d**), ReaChR-Citrine (black trace), ReaChR-Citrine F259Y (red trace) (**e**), Chrimson-EYFP (black trace) and Chrimson-EYFP Y261F (red trace) (**f**) immediately after cessation of 0.5 s illumination at a saturating light intensity of 23 mW/mm$^{-2}$ and a wavelength of **c** $\lambda = 473$ nm, **d** $\lambda = 532$ nm, **e** $\lambda = 532$ nm and **f** $\lambda = 594$ nm. **g** Typical photocurrents of Chrimson-EYFP mutants, which were measured in response to 3 ms light-pulses (23 mW/mm$^2$, $\lambda = 594$ nm). **h** For clear illustration solely the photocurrents of Chrimson-EYFP, Chrimson-EYFP Y261F/S267M (f-Chrimson-EYFP) and Chrimson-EYFP K176R/Y261F/S267M (vf-Chrimson-EYFP) are shown. Photocurrents were normalized for comparison. Scale bars: **c** 10 ms, **d**, **f** 30 ms, **e** 100 ms, **g**, **h** 20 ms

($\tau_{off} = 3.6 \pm 0.2$ ms)[6], the fastest ChR known to date. Of note, the action spectra of f-Chrimson and vf-Chrimson were not blue-shifted (Supplementary Fig. 1), thereby preserving the benefits of longer wavelength activation. Compared to Chrimson wt, the functional properties of the mutants were almost unaltered with respect to the linear voltage dependence of the photocurrents (Supplementary Fig. 2), cation permeabilities (Supplementary Table 3), the moderate slowing of the closing kinetics at positive voltages (Supplementary Fig. 3) and peak current inactivation (Supplementary Fig. 4). Of note, the measured cation permeabilities contradict a recent publication[26], but are in accordance with a previous study[6]. At a temperature of 34 °C we measured $\tau_{off}$ values of $3.2 \pm 0.2$ ms ($n = 3$) for f-Chrimson and $1.6 \pm 0.1$ ms ($n = 3$) for vf-Chrimson (Supplementary Fig. 5). Hence the ultrafast kinetics of the Chrimson mutants in principle enables

neural photostimulation in an exceptionally high frequency range of up to ~600 Hz.

**Ultrafast red-shifted optogenetics**. We heterologously expressed f-Chrimson and vf-Chrimson in primary cultures of rat hippocampal neurons by means of adeno-associated virus-mediated gene transfer (AAV2/1). Patch-clamp experiments proved a robust neuronal expression and confirmed the substantially faster kinetics of the mutants (Supplementary Table 4). The application of light pulses ($\lambda_1 = 594$ nm, $\lambda_2 = 640$ nm) triggered spiking with high reliability (Fig. 2 and Supplementary Fig. 7). The investigation of the dependence of spike probability on light pulse intensity showed that neural photostimulation via f-Chrimson (0.37–1.27 mW/mm$^2$) and vf-Chrimson (0.09–3.18 mW/mm$^2$)

### Table 1 Off-kinetics ($\tau_{off}$) of channelrhodopsin variants

| Channelrhodopsin variant | $\tau_{off}$ (ms) |
|---|---|
| ChR2 | 9.5 ± 2.8 |
| ChR2 F219Y | 5.2 ± 1.3 |
| ChR2 L132C | 16 ± 3 [a] |
| ReaChR | 361.0 ± 75.8 |
| ReaChR F259Y | 28.8 ± 3.8 |
| ReaChR L172C | 3103.7 ± 1445.2 |
| VChR1 | 119.7 ± 9.7 |
| VChR1 F214Y | 12.6 ± 1.6 |
| VChR1 L127C | 656.4 ± 129.8 |
| Chrimson | 24.6 ± 0.9 |
| Chrimson Y261F | 9.7 ± 1.5 |
| Chrimson L174C | 52.8 ± 6.0 |
| Chrimson K176R | 12.2 ± 0.8 |
| Chrimson S267M | 12.1 ± 1.5 |
| Chrimson Y268F | 11.3 ± 1.0 |
| Chrimson S267M/Y268F | 6.3 ± 1.0 |
| Chrimson Y261F/S267M | 5.7 ± 0.5 |
| Chrimson K176R/S267M/Y268F | 4.9 ± 0.5 |
| Chrimson Y261F/S267M/Y268F | 3.5 ± 0.5 |
| Chrimson K176R/Y261F/S267M | 2.7 ± 0.3 |
| Chrimson K176R/Y261F/S267M/Y268F | 2.8 ± 0.3 |

Shown are the average $\tau_{off}$ values and the corresponding standard deviations. NG cells transiently expressing channelrhodopsin variants were investigated by whole-cell patch-clamp experiments at −60 mV. The $\tau_{off}$ values were determined as described in the Methods section. ChR2 (*Chlamydomonas reinhardtii* Channelrhodopsin 2[5], $n = 3$), ChR2 L132C (CatCh[10]), ReaChR (Red-activatable Channelrhodopsin[8], $n = 3$), VChR1 (Volvox channelrhodopsin 1[7], $n = 3$), Chrimson (*Chlamydomonas noctigama* Channelrhodopsin[6], $n = 3$), Chrimson K176R (ChrimsonR[6], $n = 5$), homologous mutants to CatCh: ReaChR L172C ($n = 3$), VChR1 L127C ($n = 3$), Chrimson L174C ($n = 11$), novel mutants: Chrimson Y261F/S267M (f-Chrimson, $n = 4$), Chrimson K176R/Y261F/S267M (vf-Chrimson, $n = 7$), ChR2 F219Y ($n = 4$), ReaChR F259Y ($n = 3$), VChR1 F214Y ($n = 3$), Chrimson Y261F ($n = 7$), Chrimson S267M ($n = 5$), Chrimson Y268F ($n = 4$), Chrimson S267M/Y268F ($n = 3$), Chrimson K176R/S267M/Y268F ($n = 3$), Chrimson Y261F/S267M/Y268F ($n = 4$), Chrimson K176R/Y261F/S267M/Y268F ($n = 5$) [a] value is taken from [10]

required stronger light compared to the neural photostimulation via Chrimson wt (0.09–0.7 mW/mm$^2$) (Fig. 2e–g and Supplementary Table 4). That finding could be explained by the investigation of the light dependence of the Chrimson variants, which revealed significantly higher EC50 values for the fast Chrimson mutants (Supplementary Fig. 6). Higher EC50 values for the fast Chrimson mutants are expected, because, as described above, light sensitivity is regulated via the open time of the channel[12]. However, due to the high expression of the fast Chrimson mutants in neurons, which is demonstrated by the large current densitiy (~30 pA/pF) of their photocurrents (Supplementary Table 4) spiking with a probability of 100 % could be triggered with low intensity light pulses of 0.65 ± 0.31 mW/mm$^2$ ($n = 15$) for f-Chrimson and 1.25 ± 1.02 mW/mm$^2$ ($n = 15$) for vf-Chrimson (Supplementary Table 4). Note that potential space clamp issues might have lowered the determined current density values. The large variability of the dependence of the spike probability on the light intensity (Fig. 2e–g and Supplementary Table 4) is likely due to expression differences as well as the variability of membrane resistance, capacitance and spiking threshold of the investigated neurons.

The primary culture of rat hippocampal neurons comprises a multitude of different neuronal subtypes, most of which have a maximal firing frequency of 40–60 Hz[9]. Therefore, in most cases spike failures occurred at a frequency of 60 Hz (Fig. 2c). In single cases a frequency of 100 Hz was achieved (Fig. 2d). The investigation of neural photostimulation in the high frequency range is impeded by the heterogeneity of the primary neuronal culture. Therefore, we conducted patch-clamp experiments on parvalbumin-positive interneurons heterologously expressing vf-Chrimson. Parvalbumin-positive interneurons display a fast

spiking phenotype, and predominantly supply inhibition to the perisomatic domain of other neurons[14]. Heterologous expression of vf-Chrimson was achieved by intracerebroventricular injection of AAVs in transgenic mice that expressed tdTomato under the control of the parvalbumin promotor. Therefore, parvalbumin-positive interneurons could be identified in neocortical brain slices by their red fluorescence (Supplementary Fig. 8a).

Using current injections we determined a maximal intrinsic firing frequency of 301 ± 29 Hz ($n = 8$) for the parvalbumin-positive interneurons, as expected for the fast spiking phenotype (Fig. 3a, b)[14]. Of note vf-Chrimson enabled neural photostimulation up to the intrinsic limit of the cells with high temporal precision (Fig. 3c–e and Supplementary Fig. 8c). As demonstrated, some cells followed photostimulation up to 400 Hz (Fig. 3c, d). In 2/7 cells the occurrence of extra spikes in response to the light pulse was observed (Supplementary Fig. 8b)[9], which compromised the fidelity of neural photostimulation in those cases. We note that similar to previous work using AAV transduction and single photon stimulation[6], it was necessary to adjust irradiation intensity individually for each neuron to achieve optimal stimulation fidelity (Supplementary Fig. 9). To our knowledge these results represent the fastest light triggered spiking measured to date, and indicate that vf-Chrimson opens new possibilities for the investigation of high frequency network events, such as sharp wave-ripples[27].

**f-Chrimson is a promising candidate for hearing restoration.** Optogenetics bears great potential for improving the restoration of vision and hearing[28,29]. Future oCIs shall use tens to hundreds of microscale light sources to focally stimulate tonotopically-ordered SGNs in Rosenthal's canal (Fig. 4a)[28]. For deaf people, the lower spread of excitation from the light source in oCIs[15], promises improved frequency and intensity resolution when compared to the eCI[28]. However, much remained to be done prior to a potential clinical translation of the oCI. For example, so far cochlear optogenetics was established using blue ChRs expressed in transgenic rodents or in mice following prenatal viral-gene transfer[15].

Here, we tested the potential of f-Chrimson for optogenetic stimulation of SGNs. We established postnatal viral gene transfer by injecting AAV2/6-hSyn-f-Chrimson-EYFP into the scala tympani via the round window in 3–6-day-old mice (Fig. 4b). We readily observed photocurrents in patch-clamp recordings from isolated SGNs[30] in the second postnatal week (Fig. 4c), proving the basic functionality of f-Chrimson in the target cells. We then in depth analyzed expression and function 4–14 weeks after injection. Using confocal imaging of EYFP and parvalbumin immunofluorescence in cochlear cryosections we found a high transduction rate (near 80 %) in the injected ear, which was not significantly different between the cochlear turns (Kruskal–Wallis ANOVA followed by Dunn's test, $P > 0.05$, $n = 5$; Fig. 4d, f). SGN showed clear plasma membrane expression of f-Chrimson (insets in Fig. 4f) and survived the optogenetic manipulation as evident from the unaltered SGN density when compared to the non-injected ear (Mann–Whitney $U$ test, $P > 0.05$, $n = 5$; Fig. 4e, f). The non-injected ear showed f-Chrimson expression in less than 5% of the SGNs (Fig. 4d, f), indicating minimal spread of AAV from the injected ear likely via the cochlear aqueduct.

We then established single-channel oCI stimulation by performing a posterior tympanotomy and inserting an optical fiber (50 μm diameter) through the round window to project the light of a 594 nm laser onto the SGNs of the basal cochlear turn of young mice (2–3 months, Fig. 5a). We could readily elicit optical auditory brainstem responses (oABR, Fig. 5b, c) that differed between animals in waveform and amplitude. For comparison we

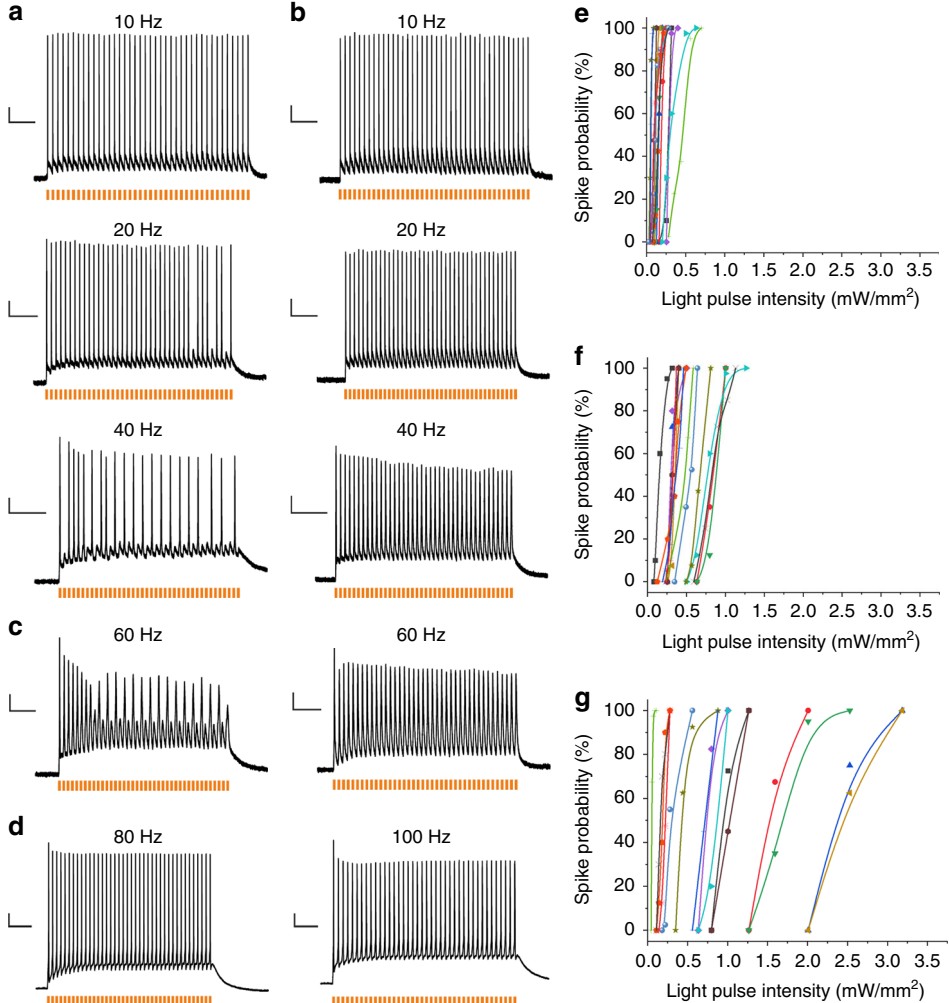

**Fig. 2** Light-induced spiking in rat hippocampal neurons. **a**–**d** Spiking traces at different light-pulse frequencies. Rat hippocampal neurons heterologously expressing Chrimson-EYFP (**a**), Chrimson-EYFP K176R/Y261F/S267M (vf-Chrimson-EYFP) (**b**) and Chrimson-EYFP Y261F/S267M (f-Chrimson-EYFP) (**c**, **d**) were investigated by whole cell patch-clamp experiments under current-clamp conditions ($\lambda = 594$ nm, pulse width = 3 ms, saturating intensity of 11–30 mW/mm$^2$). **c** Traces from two different cells at a stimulation frequency of 60 Hz. **d** Traces from one cell at stimulation frequencies of 80 Hz and 100 Hz. **e**–**g** The dependence of spike probability on light pulse intensity for Chrimson-EYFP (**e**) (15 different cells), Chrimson-EYFP Y261F/S267M (f-Chrimson-EYFP) (**f**) (15 different cells) and Chrimson-EYFP K176R/Y261F/S267M (vf-Chrimson-EYFP) (**g**) (15 different cells). The action potentials were triggered by 40 pulses ($\lambda = 594$ nm, pulse width = 3 ms, $\nu = 10$ Hz) of indicated light intensities. In order to determine the spike probability, the number of light-triggered spikes was divided by the total number of light pulses. Scale bars: $y$-axis: 10 mV, time-axis: (**a**, **b**, 10 Hz) 500 ms (**a**, **b**, 20 Hz) 300 ms (**a**, **b**, 40 Hz) 200 ms (**c**, 60 Hz) 100 ms (**d**, **80** Hz) 70 ms (**d**, **100** Hz) 50 ms

recorded acoustic auditory brainstem responses (aABRs, Fig. 5b, c lower panels) that were similar in amplitude and waveform to oABR and also varied between animals (Fig. 5b). We note that the similarity to aABRs and the shorter latency ($0.93 \pm 0.13$ ms vs. approximately 3 ms[31]) and smaller maximal amplitude ($10.7 \pm 3.80$ μV vs. approximately 1000 μV[31]) of oABRs when compared to our previous report on transgenic mice[31] indicates more specific activation of the auditory pathway in the case of postnatal AAV-injection used in present study. We then characterized the oABRs in response to different light intensities, light pulse durations and light pulse rate ($n = 5$ mice). oABR amplitude grew and oABR latency got shorter with increasing light intensity (Fig. 5c, d, g). Stimuli as weak as 0.5 mW (Fig. 5c,d, duration: 1 ms, rate: 20 Hz) and as short as 80 μs (Fig. 5e, h, rate: 20 Hz, intensity: 11 mW) were sufficient to drive oABRs. Amplitudes typically varied for changes in light intensity of more than one order (Fig. 5c, d, output dynamic range >20 dB for

oABR). oABR amplitudes declined when raising stimulus rates (Fig. 5f, i). However, f-Chrimson-mediated oABRs remained sizable up to stimulus rates of 200 Hz, suggesting high temporal fidelity of light-driven SGN firing.

Next, we used aged C57BL6/J mice (9 months-old, $n = 5$ mice) to explore the potential of oCI to restore activity in the auditory pathway of a mouse model of age-related hearing loss[32], which is a major form of hearing impairment in humans. Auditory thresholds, estimated by aABR elicited by acoustic clicks were elevated to above 50 dB (SPL) ($58 \pm 3.3$ dB SPL, Fig. 5j–l, typically 20 dB in young mice) and aABR amplitudes were reduced to 1/3 of those in young mice across all SPLs tested (Fig. 5l). oABR amplitudes measured in these aged mice were comparable to values obtained for young animals (Fig. 5l), but latencies tended to be shorter and less variable (Fig. 5g and Supplementary Fig. 10h). Interestingly, we found that light pulses as short as 40 μs were able to elicit oABRs (11 mW at 20 Hz, Supplementary

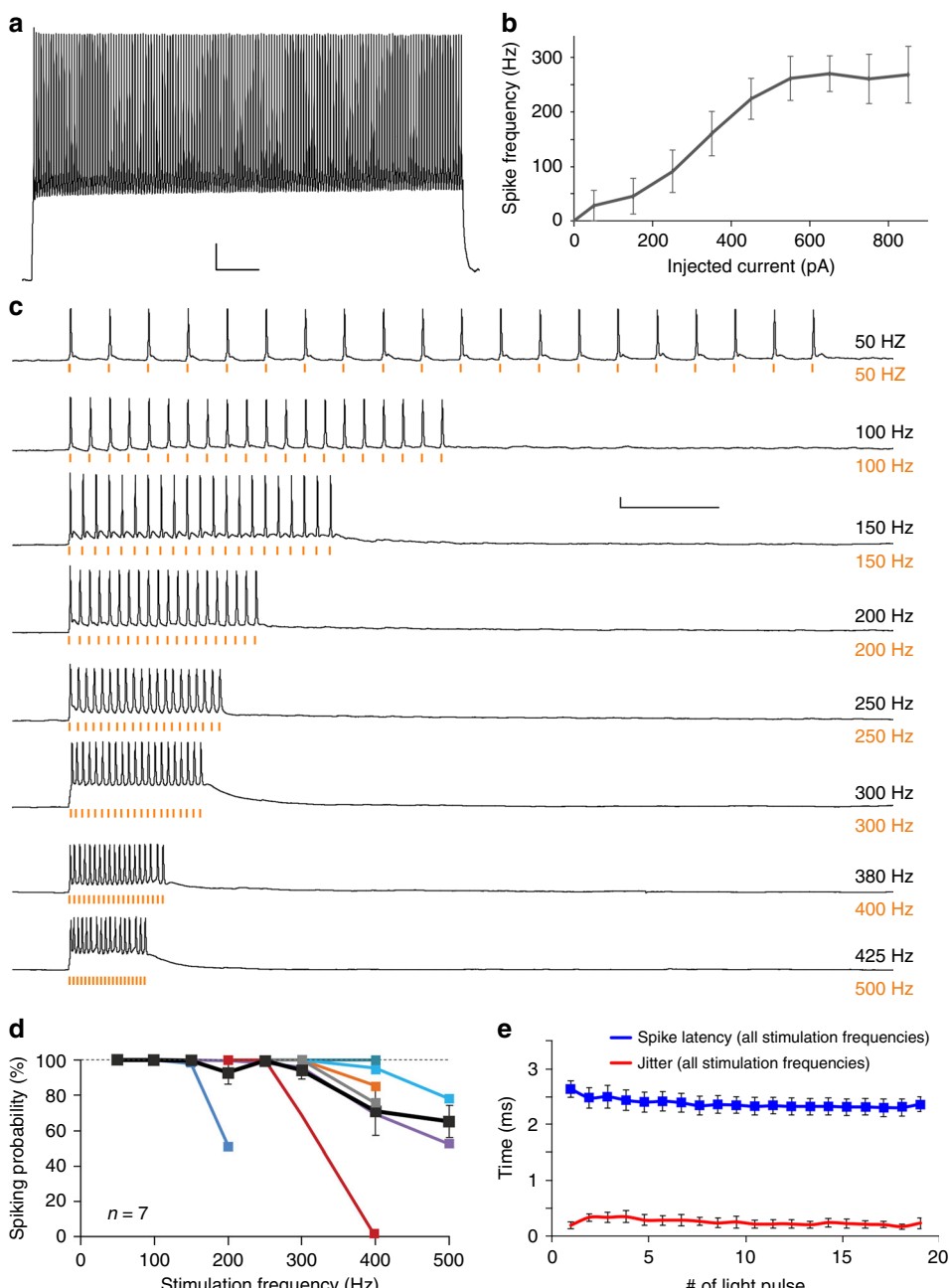

**Fig. 3** vf-Chrimson drives fast interneurons to the frequency limit. **a** Example recording of a neocortical parvalbumin-positive interneuron in an acute brain slice. Current injection (500 ms, 550 pA) elicits high frequency firing (322 Hz), consistent with the fast spiking phenotype of these interneurons. **b** When tested with constant current injection, the input–output curve of PV-interneurons plateaus at a maximum firing rate of 270 ± 33 Hz ($n = 8$). **c** Example traces of the vf Chrimson-expressing PV-interneuron from **a** activated by light pulses (565 nm, 0.5 ms) at frequencies ranging from 50–500 Hz. Note that this interneuron reliably followed frequencies of up to 400 Hz. **d** Spiking probabilities of PV-interneurons at different optical stimulation frequencies. On average (black), PV-interneurons followed stimulation up to 300 Hz reliably (94 ± 5% spiking probability), and could still encode input frequencies of up to 400 Hz with a reliability of 68 ± 16% ($n = 7$; three whole-cell, four cell-attached recordings). **e** Action potential latency (assessed at peak) and action potential jitter (s.d. of latencies) after light pulse onset for all stimulation frequencies with reliable spiking (>85%). Error bars are s.e.m. Scale bars: **a** 50 ms, 10 mV **c** 50 ms, 10 mV

Fig. 10f, i) as compared to 80 µs in young mice (Fig. 5e). Moreover, we were able to record oABRs at stimulation frequencies as high as 250 Hz (11 mW, 1 ms pulse duration), likely due to the lower average latencies found in these aged mice as compared to their younger counterparts (Supplementary Fig. 10g, j). Together the data indicate that optical activation of the auditory pathway proceeded with at least as high efficiency in aged C57BL/6J mice despite their profound age-related hearing

impairment. f-Chrimson expression levels throughout the injected cochlea were homogeneous (one-way ANOVA followed by Tukey's test, $P > 0.05$, $n = 5$). Importantly, long-term f-Chrimson expression (9 months) did not seem to decay significantly (Kruskal–Wallis ANOVA followed by Dunn's test, $P > 0.05$, $n = 5$) (Supplementary Fig. 10a–c) nor cause any significant loss of SGNs in the AAV-injected ear of these mice, when compared to the non-injected ear (t-test for comparison of

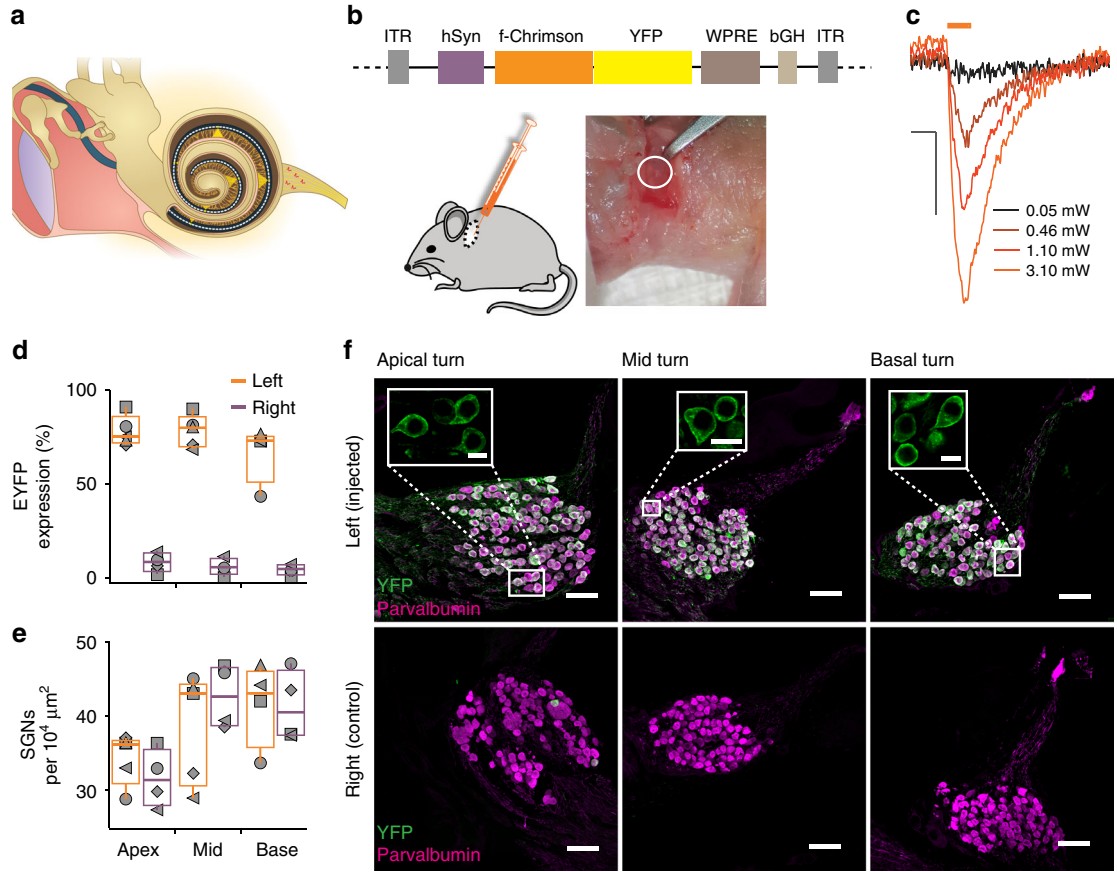

**Fig. 4** f-Chrimson expression after postnatal AAV-transduction of SGNs. **a** Scheme of the future oCI as implanted into the human ear: the oCI passes through the middle ear (limited left by ear drum and right by inner ear) near the ossicles, enters the cochlea and spirals up in scala tympani. It will likely contain tens of microscale emitters (orange spots on oCI) that stimulate (orange beams) SGNs housed in the modiolus (central compartment of the cochlea), that encode information as APs. SGNs form the auditory nerve (right) which carries the information to the brain (not displayed). **b** pAAV vector used in the study to express f-Chrimson-EYFP under the control of the hSynapsin promoter (top) upon early postnatal injection of AAV2/6 into scala tympani via a posterior tympanotomy (lower left) to expose the round window (white circle in right lower panel). **c** Photocurrents of a representative culture f-Chrimson-EYFP-positive SGN isolated from an injected ear at postnatal day 14. Light pulses of 2 ms duration were applied at the indicated intensities in the focal plane and photocurrents recorded at $-73$ mV at room temperature. Scale bar: 2 ms, 50 pA. **d** Fraction of EYFP-positive (EYFP$^+$) SGNs (identified by parvalbumin immunofluorescence, parvalbumin$^+$) and **e** density of parvalbumin$^+$ SGNs (#cells per $10^4$ μm$^2$) obtained from data as in **f**. Symbols mark results from individual animals ($n = 5$), box–whisker plots show 10th, 25th, 50th, 75th and 90th percentiles of the injected (orange) and non-injected control (magenta) cochleae (Kruskal–Wallis ANOVA, $P = 0.6538$, $H = 0.98$; post-hoc Dunn's test for comparison of expression, $P > 0.05$ for all pairwise comparisons; Mann–Whitney $U$ test for comparison of density, $L_{apex}$ vs. $R_{apex}$, $L_{mid}$ vs. $R_{mid}$, $L_{base}$ vs. $R_{base}$, $P > 0.05$ for all comparisons). **f** Projections of confocal cryosections with YFP (green) and parvalbumin (magenta) immunofluorescence of SGNs in three cochlear regions (scale bar: 50 μm). Insets (scale bar: 10 μm) show close-up images of single z-sections of the same images

cell density across cochlear turns in the injected and non-injected ear, $P > 0.05$, $n = 5$) (Supplementary Fig. 10d).

In order to scrutinize the temporal fidelity of stimulation, we turned to juxtacellular recordings from single neurons[13,33]. We established single-channel oCI stimulation via an optical fiber and targeted electrodes through a craniotomy to where the auditory nerve enters the cochlear nucleus (Fig. 6a) in order to measure the neural photoactivation. Those neurons could not be identified based on a response to acoustic stimulation, most likely due to impaired acoustic hearing following ear surgery and oCI. Therefore, we termed light-stimulated neurons putative SGNs. We found that the putative SGNs fired upon optogenetic stimulation with high temporal precision for stimulus rates of up to hundreds of Hz (Fig. 6b–e): some neurons followed stimulation to some extent even up to 1 kHz (Fig. 6b, d). The spike latency amounted to approximately 2 ms for stimulus rates of up to 400 Hz (Supplementary Fig. 11a, b), which is in agreement with the data obtained on the interneurons (Fig. 3e).

Temporal precision of firing, evaluated based on vector strength (ref. [34], see methods, Fig. 6c, d) and temporal jitter (i.e., standard deviation of spike latency across trials, Fig. 6e, Supplementary Fig. 11c) varied between the recorded neurons and, generally, was good. The vector strength declined with increasing stimulation rate up to 1 kHz. For a comparison, we re-plot the median vector strength of firing driven by transposed tones in mouse SGNs (ref. [35], Fig. 6d) used because phase-locking to pure tones is hard to achieve in the high frequency mouse cochlea[36]. Temporal jitter, evaluated for spikes occurring in the time window equal to the period stimulus, was typically below a millisecond and tended to increase when raising stimulus rates up to 300 Hz (Fig. 6e). At higher stimulus rates, the temporal jitter was higher than the values obtained for simulated Poisson spike trains (see methods, gray area, Fig. 6e), reflecting a reduced spike synchronization with the light pulses. Interestingly, the spike jitter increased significantly from 25 ms compared to the start of the stimulation (Supplementary Fig. 11c). Spike probability (Fig. 6c, d) declined

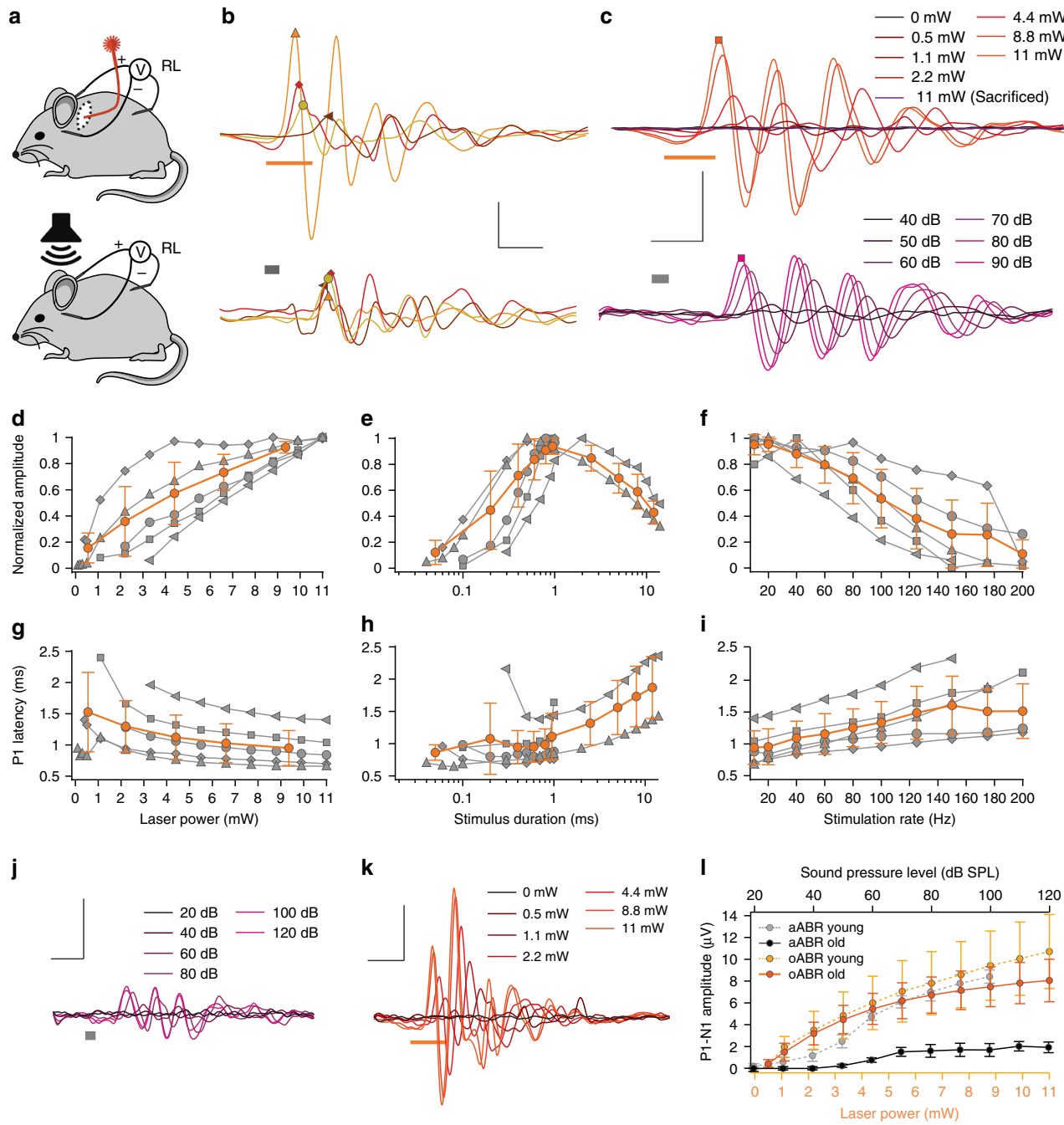

**Fig. 5** Single-channel oCIs drive oABRs in hearing and deaf mice. **a** Experimental set-up for oABR-recordings in mice: a 50 μm optical fiber coupled to a 594 nm Obis laser was implanted into scala tympani via a posterior tympanotomy and the round window. Recordings of far-field optically evoked potentials were performed by intradermal needle electrodes. For aABR recordings a free-field speaker was employed (lower panel). **b** Comparing oABRs (upper panel) and aABRs (lower panel) at strong stimulation levels for four mice (average of 1000 trials). oABRs were recorded in response to 1 ms long, 11 mW, 594 nm laser pulse at 10 Hz, aABRs of the same mice in response to 80 dB (SPL peak equivalent) clicks. Bars indicate the stimulus timing. **c** oABRs (upper panel, 594 nm, 1 ms at 10 s⁻¹) and aABRs (lower panel, clicks at 10 s⁻¹, values in SPL [peak equivalent]) recorded from an exemplary AAV-injected mouse at increasing stimulus intensities. **d–f** Normalized P1-N1-amplitude as a function of laser intensity (**d** 1 ms at 20 Hz), pulse duration (**e** 11 mW at 20 Hz), and stimulus rate (**f** 11 mW, 1 ms). Group average (lines) and s.d. (error bars) are shown in orange (same for **g–i**). **g–i** P1-latency as a function of laser intensity (**g** as in **d**), duration (**h** as in **e**), and rate (**i** as in **f**). **j** Exemplary aABR recordings done as in **a–c** using a 9 months-old mouse (following postnatal AAV-Chrimson-EYFP injection: elevated acoustic thresholds (around 60 dB [SPL], compare to **c**). **k** oABR recordings done as in **a–c** in the same mouse as in **j**, using 1 ms long laser pulses: thresholds similar to injected mice at 2–3 months of age (around 1 mW, compare to **c**). **l** P1-N1-amplitude of oABR (orange) and P1-N1-amplitude of aABR (gray) as function of stimulus intensity in young (2–3 months-old) and old (9 months-old) mice (n = 5 for each group, means (lines) ± s.e.m. (error bars) are shown. Symbols in **d–i** mark results from individual animals. Scale bars (**b**, **c**, **j**, **k**): 1 ms, 5 μV

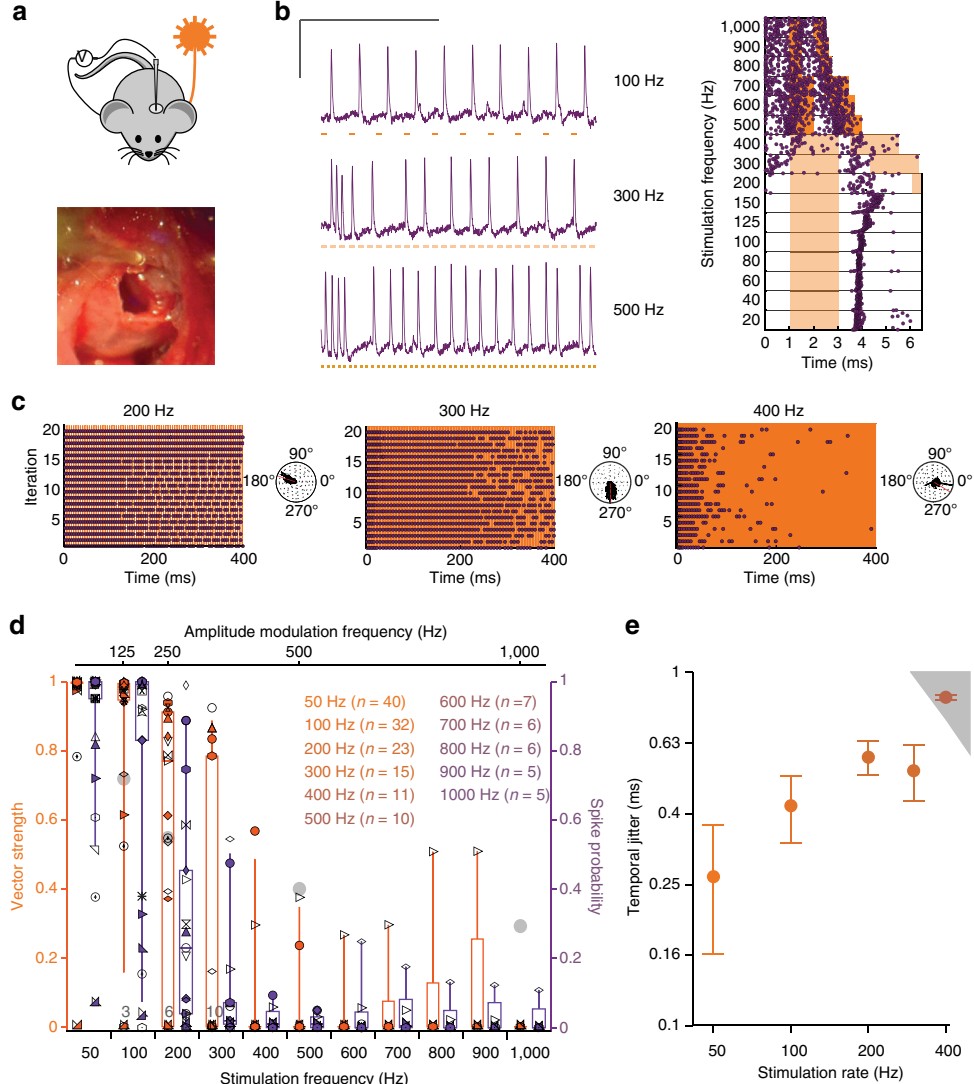

**Fig. 6** f-Chrimson enables SGNs spiking at near physiological rates. **a** Experimental set-up for recording optogenetic responses of SGNs in mice: a 50 μm optical fiber coupled to a 594 nm laser was implanted into scala tympani via the round window (lower panel, see cylindrical structure in the upper half) and microelectrodes were advanced into the cochlear nucleus via a craniotomy (upper panel). **b** Exemplary spikes of a neuron (1 ms, 5.5 mW for 100, 300 Hz; 11 mW for 500 Hz). Raster plot (right panel): spike times in response to laser pulses (orange bars: 2 ms @5.5 mW for 20-400 Hz, 1 ms @11 mW for 500-700 Hz and above: 0.5 ms @11 mW): spikes cluster in time for stimulus rates up to hundreds of Hz, temporal jitter increases with stimulation rates. Scale bar: 50 ms, 2 mV. **c** Activity of an exemplary neuron in response to 900 ms trains of laser pulses (1 ms) at three different rates leaving an inter-train recovery time of 100 ms (first 400 ms are shown and analyzed). Panels to the right side of raster plots show polar plots: synchronicity and probability of firing decay with increasing stimulus frequency. Spike probability 200 Hz: 0.8, 300 Hz: 0.33, 400 Hz: 0.04. Vector strength 200 Hz: 0.92, 300 Hz: 0.83, 400 Hz: 0.57 (Rayleigh-test: $P < 0.001$ in all cases). **d** Box-whisker plots showing 10th, 25th, 50th, 75th and 90th percentiles of the vector strength (orange) and spike probability (purple) of 40 units from five mice, stimulated at different rates as described for **c**. Symbols represent values from every unit. Gray circles are means of vector strength of SGNs in wild-type mice found with transposed tones at the characteristic frequency at 30 dB relative to spike threshold[36], for comparison. Numbers at the bottom of the graph indicate number of units clustered below them. **e** Temporal jitter of spikes across stimulation rates 50–400 Hz. Gray area represents the hazard function obtained in response to simulated Poisson spike trains. Data points show mean (lines) ± s.e.m. (error bars). Number of units included for each stimulation frequency (color coded) is shown

as the rate of stimulation increased, indicating that optogenetic coding by individual SGNs becomes less reliable as stimulus rate rises. This, however, is likely compensated at the population level, as several SGNs jointly encode information from each place of the tonotopic map[13].

## Discussion
As demonstrated, the investigation of the molecular properties of microbial-type rhodopsins is essential for the development of variants with superior properties for particular optogenetic applications. Our study reveals the critical role of helix F for the closing kinetics of various ChRs. Using site-directed mutagenesis we generated fast variants of four different ChRs, which, together, cover photoactivation over the visible spectrum. We deem the fast (f−) and very fast (vf−) Chrimson of particular interest to the neurosciences because of their red-shifted action spectrum and high membrane expression. Our analysis of fast spiking inter-neurons of the cerebral cortex demonstrated that they enable the remote optical control of even the fastest neurons at their intrinsic physiological limits. Finally, we show that f-Chrimson is

a promising candidate for future clinical optogenetic restoration of sensory function.

Channel opening/closing of ChR2 is based on a concerted movement of helices B, F and G[23,24]. Interestingly, we discovered a major impact of the interaction between the moving helix F and the virtually immobile helix C on ChR kinetics. High-resolution structures of the investigated ChRs are not available. However, the information of the high-resolution structure of the C1C2 ChR chimera in combination with the light-induced helix movement studies by electron spin resonance and the low-resolution structure by 2D cryoelectron microscopy allowed us to identify the crucial position F219 in helix F for the construction of a faster ChR2 mutant. This position is also conserved in ReaChR and VCR1. Analogous mutations lead to accelerated kinetics of channel closing. Of note, the surprisingly slow reacting Chrimson has already a tyrosine on this position (Fig. 1a, b). We suspected that the mutation of this tyrosine influences channel closing kinetics. Indeed, the mutation to a phenylalanine in the analogous position (Y261F) results to faster channel closing in Chrimson. Further inspection of the alignment of helix F shows at positions M225 and F226 in ChR2 the same analogy for VCR1 and ReachR but not for Chrimson. For Chrimson, mutations on these positions (S267M, Y268F) in addition to the Y261F mutation result in the ultrafast switching behavior (Fig. 1g,h). The surprising and peculiar phenotype of the back mutations in Chrimson is hard to explain without a high resolution structure of the protein.

This study achieved two important breakthroughs towards developing cochlear optogenetics for auditory research and future improved hearing restoration. First, we managed to achieve efficient, non-traumatic and neuron-specific expression of f-Chrimson in SGNs using postnatal AAV-injection into scala tympani through the round window. We found near 80% of the SGNs of the injected ear to express f-Chrimson at high levels and mostly in the plasma membrane of somas and neurites, which persists for at least 9 months after injection. These transduction rates were much higher than those achieved with transuterine injection of AAV2/6-hSyn-CatCh-YFP[15] and unlike there, independent from tonotopic position. We consider the minute transduction of the non-injected ear to reflect viral spread through the cerebrospinal fluid space, which calls for further optimization of the injection protocol[37]. Importantly, we did not find evidence for neuron loss even at 9 months after injection and we expect little, if any, risk of phototoxicity given the red-shifted action spectrum of f-Chrimson.

Secondly, using f-Chrimson, we overcame the likely biggest roadblock of current ChR2-based cochlear optogenetics: low temporal bandwidth of optical coding (<80 Hz)[15]. We found robust and fast photocurrents in cultured isolated f-Chrimson-positive SGNs. In vivo, fiber-based stimulation resembling single-channel oCI elicited activation of the auditory pathway in hearing and deaf mice. Using far-field neural population responses (oABR) as a readout we found low thresholds for radiant flux and energy (<0.5 mW, <0.5 µJ) as well as duration (<100 µs) and a wide dynamic range of coding (more than 20 dB, no saturation for most animals at maximal stimulation). This brings the oCI closer to the currently used eCI stimulation parameters (0.2 µJ and 80 µs per pulse)[38] and exceeds the eCI output dynamic range (<10 dB)[17]. Both, recordings of oABR and of firing in single putative SGNs indicated that f-Chrimson mediated oCI-enabled responses to follow at least 200 Hz, which corresponds to physiological steady-state firing rates of SGNs[13]. In fact, we found some neurons to follow stimulation to several hundreds of Hz nearly mimicking sound-evoked SGN activity. The closing kinetics of f-Chrimson and the resulting relative refractoriness probably also limits the temporal precision of f-Chrimson-

mediated SGN firing. At 500 and 1000 Hz, vector strength, a measure commonly used to analyze the extent of phase-locking in SGNs[39], was lower for f-Chrimson-mediated optogenetic stimulation than for mouse SGN firing with transposed tones[35]. We note that the SGNs recorded in the present study typically did not fire spontaneously probably due to the ear surgery. Besides the lack of spontaneous firing, the short (1 ms) and pulsatile optogenetic stimulation typically evoking a single spike likely explain why vector strength tended to be higher for low stimulus rates when compared to transposed tones, for which several spikes were generated per stimulus cycle. Moreover, vector strength and temporal jitter of f-Chrimson-mediated SGN firing in mice indicate a lower temporal precision than that of acoustic hearing and electric stimulation in species with prominent phase-locking of SGN firing[33,39]. Nonetheless, we reason that, even if the limited probability and temporal precision of single SGN firing for optogenetic stimulation at 100–500 Hz translated to species other the mouse, this will not impede the coding at the level of the auditory nerve population. Hence, we conclude that f-Chrimson is a good candidate opsin for the oCI. In fact, higher temporal jitter in response to optogenetic than electrical stimulation might render unnecessary the very high stimulation rates employed in eCI to avoid overly synchronized activity in the auditory nerve[17]. The favorable properties of the novel Chrimson mutants also facilitate multiple applications in basic neurosciences and in sensory restoration, such as the recovery of vision[29].

## Methods

**Molecular biology.** The humanized DNA sequence, coding for the red light activated ChR Chrimson from *Chlamydomonas noctigama* (accession number: KF992060), either C-terminally fused to EYFP or without a fluorescent tag, was cloned into the mammalian expression vector pcDNA3.1(−) (Invitrogen, Carlsbad, USA). The mutations L174K, K176R (ChrimsonR), Y261F, S267M and Y268F as well as combinations of aforementioned mutations (Table 1) were created by site-directed mutagenesis. Chrimson-EYFP wt and Chrimson-EYFP Y261F/S267M (f-Chrimson-EYFP) were subcloned into the *Xenopus laevis* oocyte expression vector pTLN[40].

We also cloned the humanized DNA sequences coding for ChR2 (C-terminally truncated variant Chop2-315 of ChR2 from *Chlamydomonas reinhardtii*, accession number: AF461397), for Volvox ChR 1 (VChR1, accession number: EU622855) and for the chimera ReaChR (ChR1/VChR1/VChR2, Red-activatable ChR, accession number: KF448069) into the mammalian expression vector pcDNA3.1 (−) (Invitrogen, Carlsbad, USA). Thereby ChR2 and VChR1 were C-terminally fused to EYFP and ReaChR was C-terminally fused to Citrine. The mutants ChR2-EYFP F219Y, VChR1-EYFP F214Y, VChR1-EYFP L127C, ReaChR-Citrine F259Y and ReaChR-Citrine L172C were created by site-directed mutagenesis. The related primer sequences were shown in Supplementary Tables 5 and 6.

**NG108-15 cell culture and transfection.** NG108-15 cells (ATCC, HB-12377™, Manassas, USA) were cultured at 37 °C and 5% CO$_2$ in DMEM (Sigma, St. Louis, USA) supplemented with 10% fetal calf serum (Sigma, St. Louis, USA), and 5 % penicillin/streptomycin (Sigma, St. Louis, USA). One day prior to transient transfections the NG108-15 cells were seeded on 24-well plates. Two to three days prior to their electrophysiological characterization by patch-clamp experiments the NG108-15 cells were transiently transfected with pcDNA3.1(−) derivatives carrying aforementioned ChRs and ChR mutants using Lipofectamine 2000 (Invitrogen, Carlsbad, USA) or Lipofectamine LTX (Invitrogen, Carlsbad, USA). Cells were tested for mycoplasma contamination using specific primers. No method of cell line authentication was used.

**Expression of Chrimson variants in *Xenopus laevis* oocytes.** *Xenopus laevis* oocytes were injected with 50 ng of in vitro-transcribed cRNA (Thermo Fisher Scientific, Waltham, USA), coding for Chrimson-EYFP wt and Chrimson-EYFP Y261F/S267M (f-Chrimson-EYFP). After cRNA injection the *Xenopus laevis* oocytes were incubated at 16 °C in an 1 µM all-trans retinal containing Barth's solution (88 mM NaCl, 1 mM KCl, 0.33 mM Ca(NO$_3$)$_2$, 0.41 CaCl$_2$, 0.82 MgSO$_4$, 2.4 mM NaHCO$_3$, 10 mM HEPES, pH 7.4 supplemented with 50 mg/l gentamycin, 67 mg/l penicillin and 100 mg/l streptomycin) for 4–5 days.

**Electrophysiological recordings on *Xenopus laevis* oocytes.** The *Xenopus laevis* oocytes heterologously expressing the Chrimson variants were investigated by the two electrode voltage-clamp technique[5,41]. Photocurrents were measured in response to 500 ms light pulses with a wavelength of $\lambda = 590$ nm using the LED

OEM module (Omikron, Rodgau-Dudenhofen, Germany) focused into a 2 mm optic fiber.

In order to assess the permeability of potassium ions relative to the permeability of sodium ions ($P_K/P_{Na}$), we measured photocurrents at voltages ranging from −120 to +40 mV in 20-mV steps. The $P_K/P_{Na}$ ratio was determined from the difference of the reversal potentials of the photocurrents when replacing 90 mM NaCl, 2 mM MgCl$_2$, 5 mM MOPS/TRIS pH 9 for 90 mM KCl, 2 mM MgCl$_2$, 5 mM MOPS/TRIS pH 9[5]. The relative proton permeability was calculated from the photocurrent reversal potential in buffer containing 90 mM NMG, 5 mM KCl, 2 mM MgCl$_2$, 5 mM MOPS/TRIS pH 9 using the Goldmann–Hodgkin–Katz equation[25] and assuming a cytoplasmic potassium concentration of 100 mM and an intracellular pH of 7.3[5].

**Electrophysiological recordings on NG108-15 cells**. For the electrophysiological characterization of mutant ChRs whole cell patch-clamp were performed under voltage clamp conditions[42] using the Axopatch 200B amplifier (Axon Instruments, Union City, USA) and the DigiData 1322A interface (Axon Instruments, Union City, USA). Patch pipettes with resistances of 2–5 mΩ were fabricated from thin-walled borosilicate glass on a horizontal puller (Model P-1000, Sutter Instruments, Novato, USA). The series resistance was <10 MΩ and the input resistance ranged from 1.1 to 4.6 GΩ. The mean capacitance of the measured cells was 34.6 ± 24.3 pF ($n$ = 61). If not stated differently the pipette solution contained 110 mM NaCl, 2 mM MgCl$_2$, 10 mM EGTA, 10 mM HEPES, pH 7.4 and the bath solution contained 140 mM NaCl, 2 mM CaCl$_2$, 2 mM MgCl$_2$, 10 mM HEPES, pH 7.4.

In order to assess the permeability of calcium ions relative to the permeability of sodium ions ($P_{Ca}/P_{Na}$), we measured photocurrent–voltage relationships and determined the reversal potential. The intracellular solution contained 110 mM NaCl, 10 mM EGTA, 2 mM MgCl$_2$ and 10 mM Tris (pH = 7.4) and the extracellular solution contained 140 mM NaCl, 2 mM MgCl$_2$ and 10 mM Tris (pH = 9). For the determination of the $P_{Ca}/P_{Na}$ values, external 140 mM NaCl was exchanged with 90 mM CaCl$_2$. Permeability ratios were calculated according to the Goldman–Hodgkin–Katz equation[25].

For determination and comparison of the off-kinetics and current densities, NG108-15 cells heterologously expressing aforementioned ChRs and channelrhodopsin mutants were investigated at a membrane potential of −60 mV. Photocurrents were measured in response to 3 or 500 ms light pulses with a saturating intensity of 23 mW/mm$^2$ using diode-pumped solid-state lasers ($\lambda$ = 473 nm for ChR2 variants, $\lambda$ = 532 nm for VChR1 and ReaChR variants, $\lambda$ = 594 nm for Chrimson variants) focused into a 400-μm optic fiber. Light pulses were applied by a fast computer-controlled shutter (Uniblitz LS6ZM2, Vincent Associates, Rochester, USA).

The current density ($J_{-60\,mV}$) was determined by dividing the stationary current in response to a 500 ms light pulse with a saturating intensity of 23 mW/mm$^2$ by the capacitance of the cell. In order to avoid an experimental bias, the NG108-15 cells for the electrophysiological recordings were chosen independent of the brightness of their EYFP fluorescence. The $\tau_{off}$ value was determined by a fit of the decaying photocurrent to a monoexponential function. In order to investigate the dependence of the off-kinetics on the membrane potential $\tau_{off}$ values were determined at membrane potentials ranging from −120 to +60 mV.

If not stated differently the off-kinetics was determined at room temperature (297 K). The temperature dependence of the off-kinetics of Chrimson-EYFP wt and Chrimson-EYFP K176R/Y261F/S267M (vf-Chrimson-EYFP) was investigated at temperatures ranging from 284 to 307 K. Photocurrents recorded at a temperature of 307 K were measured in response to 7 ns light-pulses with a wavelength of 594 nm in order to avoid tampering of the off-kinetics due to the opening/closing time of the shutter (700 μs). The ns light pulses were generated with the Opolette 355 tunable laser system (Opotek Inc, Carlsbad, USA). Thereby the pulse energy was set to value of 10$^{19}$ photons/m$^2$.

The Opolette 355 tunable laser system was further used for the measurement of the action spectra of the Chrimson variants. For the recordings the pulse energies at the different wavelengths were set to values which corresponded to equal photon counts of 10$^{18}$ photons/m$^2$ for Chrimson wt and 10$^{19}$ photons/m$^2$ for the Chrimson mutants.

**Hippocampal neuron culture**. Hippocampi were isolated from postnatal P1 Sprague–Dawley rats and treated with papain (20 U ml$^{-1}$) for 20 min at 37 °C (by the lab of Dr. Erin Schuman, MPI of Brain Research, Frankfurt). The hippocampi were washed with DMEM high glucose (Sigma-Aldrich, St. Louis, USA) supplemented with 10% fetal bovine serum and titrated in a small volume of this solution. Approximately 96,000 cells were plated on poly-D-lysine/laminin coated glass cover slips in 24-well plates. After 3 h the plating medium was replaced by culture medium containing Neurobasal A (Thermo Fisher Scientific, Waltham, USA) supplemented with 2% B-27 supplement (Thermo Fisher Scientific, Waltham, USA), and 2 mM Glutamax (Thermo Fisher Scientific, Waltham, USA).

**Adeno-associated virus (AAV2/1) transduction**. rAAV2/1 virus was prepared by the lab of Botond Roska, FMI, Basel using a pAAV2 vector with a human synapsin promoter[8] containing Chrimson, Chrimson-EYFP, Chrimson K176R,

Chrimson-EYFP K176R, Chrimson Y261F/S267M (f-Chrimson), Chrimson-EYFP Y261F/S267M (f-Chrimson-EYFP), Chrimson K176R/Y261F/S267M (vf-Chrimson) and Chrimson-EYFP K176R/Y261F/S267M (vf-Chrimson-EYFP). The virus titer was nominally $1 \times 10^{12}$–$1 \times 10^{13}$ GC/ml. Briefly $1 \times 10^9$ genome copies/ml (GC/ml) of rAAV2/1 virus was added to each well 4–9 days after plating. Expression became visible 5 days post-transduction. The electrophysiological measurements were performed 13–21 days after transduction. No neurotoxicity was observed for the lifetime of the culture (~5 weeks). No all-trans retinal was added to the culture medium or recording medium for any of the experiments described here.

**Electrophysiological recordings on hippocampal neurons**. For whole-cell recordings in cultured hippocampal neurons, patch pipettes with resistances of 3–8 MΩ were filled with 129 mM potassium gluconate, 10 mM HEPES, 10 mM KCl, 4 mM MgATP and 0.3 mM Na$_3$GTP, titrated to pH 7.2. Tyrode's solution was used as the extracellular solution (125 mM NaCl, 2 mM KCl, 2 mM CaCl$_2$, 1 mM MgCl$_2$, 30 mM glucose and 25 mM HEPES, titrated to pH 7.4). The series resistance was <10 MΩ and the input resistance ranged from 0.7 to 3.5 GΩ. The mean capacitance of the measured cells was 35.4 ± 12.4 pF ($n$ = 31). In order to avoid an experimental bias in cell selection, the neurons for the electrophysiological recordings were selected independent of the brightness of their EYFP fluorescence. Recordings were conducted in the presence of the excitatory synaptic transmission blockers, 1,2,3,4-tetrahydro-6-nitro-2,3-dioxo-benzo[f]quinoxaline-7-sulfonamide (NBQX, 10 μM, Sigma-Aldrich, St. Louis, USA) and D(−)-2-Amino-5-phosphonopentanoic acid (AP-5, 50 μM, Sigma-Aldrich, St. Louis, USA). For determination of $\tau_{off}$ and $J_{-70\,mV}$ measurements were conducted in the presence of 1 μM TTX (Sigma-Aldrich, St. Louis, USA) in addition. Electrophysiological signals were amplified using an Axopatch 200B amplifier (Axon Instruments, Union City, USA), filtered at 10 kHz, digitized with an Axon Digidata 1322 A (50 kHz) and acquired and analyzed using pClamp9 software (Axon Instruments, Union City, USA).

The light pulses had a pulse width of 3 ms, a wavelength of $\lambda$ = 594 nm and a saturating intensity of 11–30 mW/mm$^2$. The $\tau_{off}$ value was determined by a fit of the decaying photocurrent to a monoexponential function. The current density ($J_{-70\,mV}$) was determined by dividing the stationary current in response to a 500 ms light pulse with a saturating intensity of 20–40 mW/mm$^2$ and a wavelength of 594 nm by the capacitance of the cell. In order to determine the lowest light intensity required to induce action potentials with a probability of 100% ($J_{100}$), 40 pulses ($\lambda$ = 594 nm, pulse width = 3 ms, $v$ = 10 Hz) of varying light intensities were applied. The spike probability was calculated by dividing the number of light-triggered spikes by the total number of light pulses.

**Animals for recordings on parvalbumin-positive interneurons**. Experimental mice were obtained from a cross of PV-ires-cre[43] and conditional tdTomato animals Ai9, male and female, 4–12 weeks[44]. Mice were maintained in a 12 h light/dark cycle, with access to food and water ad libitum. All animal procedures were performed in accordance with institutional guidelines and were approved by the Regierungspräsidium Darmstadt.

**Intracerebroventricular injections**. Prior to pup injections[45], the dam was habituated to the experimenter and the experimental room. Newborn mice (P2) were anesthetized using isoflurane (2–3%), and placed on a light source to reveal skull structures. Injections of 2 μl of AAV2/1-hSyn-vf-Chrimson-EYFP were performed into the right ventricle using a glass pipette (coordinates from bregma; rostral 0.75 mm, lateral 0.25 mm and ventral 2 mm). After injection pups were recovered for 5 min in a pre-warmed container with homecage-bedding before being placed back in the home cage.

**Patch-clamp recordings on parvalbumin-positive interneurons**. Coronal brain slices were prepared from 2–6-week old PV-tdTomato mice that had been injected with 2 μl of AAV2/1-hSyn-vf-Chrimson-EYFP at postnatal day 2. Animals were anesthetized with isoflurane (3% in oxygen), decapitated and the brain was dissected in ice-cold artificial cerebrospinal fluid (ACSF, containing in mM: 125 NaCl, 3 KCl, 2 CaCl$_2$, 1 MgCl$_2$, 26 NaHCO$_3$, 10 glucose), and sliced (325-μm thick) on a vibrating microtom (VT1200S; Wetzlar, Germany) at 4 °C. Slices were recovered for 60 min at room temperature in a submersion chamber containing ACSF equilibrated with 95% O$_2$/5% CO$_2$. Slices were next transferred to the submersion chamber of an upright microscope (Scientifica), and continuously superfused with ACSF additionally containing 1-μM DNQX, 40-μM AP5 and 1-μM bicuculline at 33 °C. Parvalbumin-positive interneurons were identified in layer 2/3 of neocortex with a combination of infrared and fluorescence video microscopy under a 40× objective (Olympus). Patch-clamp electrodes (8–12 MΩ) were pulled from borosilicate glass and filled with an intracellular solution consisting of (in mM): 140 potassium–gluconate, 10 HEPES, 4 phosphocreatineNa$_2$, 4 Mg-ATP, 0.4 Na-GTP, 10 KCl (pH adjusted to 7.25 with KOH, ~280–300 mOsm). Data were acquired with a Multiclamp700B amplifier and pClamp 10.5 software (Axon Instruments). Optogenetically evoked action potentials were recorded in parvalbumin-positive interneurons in loose-seal cell-attached ($n$ = 4) or whole-cell current-clamp mode ($n$ = 3). In addition, another five parvalbumin-positive interneurons were recorded for the input–output curves presented in Fig. 3b. Data were filtered at 20 kHz and

sampled at 50 kHz. Spiking patterns were assessed with depolarizing current steps in eight whole-cell recordings, and displayed the fast-spiking phenotype expected for PV-interneurons ($n = 8$, maximal firing frequency $301 \pm 29$ Hz). Optical stimulation was performed through the objective by an LED (coolLED) coupled to the microscope. Pulse width was 0.25–1 ms, irradiance ranged from 1–10 mW/mm$^2$, and was adjusted individually for every neuron to cause reliable firing (Fig. 3c: 50 Hz, pulsewidth = 0.5 ms, light-density = 2.6 mW/mm$^2$; 100 Hz, pulsewidth = 0.5 ms, light-density = 2.6 mW/mm$^2$; 150 Hz, pulsewidth = 0.5 ms, light-density = 4.9 mW/mm$^2$; 200 Hz, pulsewidth = 0.5 ms, light-density = 4.9 mW/mm2; 250 Hz, pulsewidth = 0.5 ms, light-density = 4.9 mW/mm$^2$; 300 Hz, pulsewidth = 0.5 ms, light-density = 4.9 mW/mm$^2$; 400 Hz, pulsewidth = 0.5 ms, light-density = 4.9 mW/mm$^2$; 500 Hz, pulsewidth = 0.5 ms, light-density = 8 mW/mm$^2$). Data were analyzed using clampfit and excel software. For calculation of latency and jitter, the time of action potential peak was used. We note that the apparent action potential threshold defined as the voltage at which the first temporal derivative crosses a threshold of 40 V/s is more hyperpolarized for optogentically evoked action potentials ($-57.00 \pm 1.85$ mV) compared to action potentials during DC current injections ($-43.19 \pm 2.97$ mV, $p < 0.001$, unpaired, two-tailed $t$-test). Statistics were done using non-parametric Friedmann test followed by a post-hoc Dunn's test (Prism, GraphPad Sofware Inc., La Jolla, USA).

**Cloning for AAV2/6 production.** pcDNA3.1(−)_f-Chrimson_EYFP was used as a starting material for cloning pAAV_hSyn_f-Chrimson_EYFP. The sequence of f-Chrimson_EYFP was amplified by means of a classical PCR. The resulting PCR fragment was then digested with BamHI/HindIII (Thermo Scientific, MA, USA), gel extracted (GeneJET Gel Extraction Kit, Thermo Scientific, MA, USA) and further used for ligation. At the same time the plasmid pAAV_hSyn_Chronos_GFP (Addgene, plasmid nr. 59170) was also digested using restriction enzymes BamHI/HindIII and used as a backbone plasmid. All obtained ligation products were further tested by means of colony PCR and finally sequenced by an external company. The final product was then sent to the University of North Carolina Vector Core (Chapel Hill, NC, USA), and used to produce AAV2/6.

**Postnatal AAV injection into the cochlea.** All experiments were done in compliance with the German national animal care guidelines and were approved by the board for animal welfare of the University Medical Center Göttingen and the animal welfare office of the state of Lower Saxony. The calculation of animal number was performed prior to starting experiments. We planned to use the Wilcoxon Rank Sum Test and an error probability alpha smaller than 0.05, a power (1-beta) of 0.95 and effect size depending on the precise experimental protocol.

Postnatal AAV-injection into scala tympani of the left ear via the round window[46] was performed at p3-p6 on C57BL/6 wild-type mice, using AAV2/6 and the human synapsin promoter to drive transgenic expression of f-Chrimson-YFP in SGNs. In brief, under general isoflurane anesthesia and local analgesia achieved by means of xylocaine, the left ear was approached via a dorsal incision and the round window membrane was identified and gently punctured using a borosilicate capillary pipette that was kept in place to inject approximately $5 \times 10^9$ viral genomes. After virus application, the tissue above the injection site was repositioned and the wound was sutured and buprenorphine (0.1 mg/kg) was applied as pain reliever. Recovery of the animals was then daily tracked. Mice were randomly selected for injection in all experiments. No blinding was possible since injections have to be performed in the left ear leaving the right ear as an internal control. Hence, surgery prior to stimulation needed to be done in the injected ear. Animals were then kept in a 12 h light/dark cycle, with access to food and water *ad libitum*.

**Immunostaining and imaging of cochlear cryosections.** Cochleae were fixed with 4% paraformaldehyde in phosphate buffered saline for 1 h. Cochleae were then cryosectioned following 0.12 M EDTA decalcification. After incubation of sections for 1 h in goat serum dilution buffer (16% normal goat serum, 450 mM NaCl, 0.6% Triton X-100, 20 mM phosphate buffer, pH 7.4), primary antibodies were applied for 1 h at room temperature. The following antibodies were used: chicken anti-GFP (catalog number: ab13970, dilution 1:500) (Abcam, Cambridge, United Kingdom), guinea pig anti-parvalbumin (catalog number: 195004, dilution 1:300) (Synaptic Systems, Göttingen, Germany). The following secondary AlexaFluor-labeled antibodies were applied for 1 h at room temperature: goat anti-chicken 488 IgG (H + L), catalog number: A-11039, dilution 1:200 (Thermo Scientific, MA, USA); goat-anti guinea pig 568 IgG (H + L), catalog number A1107, dilution 1:200 (Thermo Scientific, MA, USA). Confocal images were collected using a SP5 microscope (Leica, Hamburg, Germany) and processed in ImageJ (NIH, Bethesda, MD, USA). Expression was considered positive when EYFP fluorescence in a given cell (marked by parvalbumin) was found to be higher than 3 SD above the background fluorescence of the tissue.

**Animal surgery for recordings on the auditory pathway.** Mice were anesthetized with i.p. administration of a mixture of xylazine (5 mg/kg) and urethane (1.32 mg/kg) while analgesia was achieved with buprenorphine. The core temperature was maintained constant at 37 °C using a custom-designed heat plate on a vibration isolation table in a sound-proof chamber (IAC GmbH, Niederkrüchten, Germany).

For auditory nerve recordings, a tracheostomy was performed before the animals were positioned in a custom-designed stereotactic head holder. Pinnae were removed, scalp reflected, portions of the lateral interparietal and of the left occipital bone removed, and a partial cerebellar aspiration performed to expose the surface of the cochlear nucleus.

**Optical stimulation in vivo.** The left bulla was reached using a retroauricular approach and opened to expose the cochlea. A 50 μm optical fiber coupled to a 594-nm laser (OBIS LS OPSL, 100 mW, Coherent Inc., Santa Clara, CA, USA) was inserted into the cochlea via the round window. Radiant flux was calibrated with a laser power meter (LaserCheck; Coherent Inc., Santa Clara, CA, USA).

**SGN culture and patch-clamp recordings.** On postnatal day 12–14, SGNs of injected mice were isolated, cultured and patch-clamped[30]. In brief, we patch-clamped EYFP-positive SGNs using an EPC-10 amplifier controlled by Patchmaster software (HEKA electronics, Lambrecht, Germany) and employing potassium-gluconate based intracellular solution (in mM: 130 K-gluconate, 5 KCl, 1 EGTA, 2 MgATP, 2 Na$_2$ATP, 0.3 MgGTP, 10 KOH-HEPES, 10Na$_2$Phosphocreatinine) and an extracellular solution containing (in mM: 145 NaCl, 4 KCl, 1 MgCl$_2$, 1.3 CaCl, 10 NaOH-HEPES. A 594-nm laser (OBIS LS OPSL, 100 mW, Coherent Inc., Santa Clara, CA, USA) was coupled into a Nikon Eclipse inverted microscope and radiant flux was calibrated using a powermeter.

**Auditory brainstem responses.** For stimulus generation and presentation, data acquisition, and off-line analysis, we used a NI System (National Instruments, Austin, TX, USA) and custom-written MATLAB software (The MathWorks, Inc., Natick, MA, USA). Optically evoked ABRs (oABRs) and acoustically evoked ABRs (aABRs) were recorded by needle electrodes underneath the pinna, on the vertex, and on the back near the legs. The difference potential between vertex and mastoid subdermal needles was amplified using a custom-designed amplifier, sampled at a rate of 50 kHz for 20 ms, filtered (300–3000 Hz) and averaged across 1000 and 500 presentations (for oABRs and aABRs, respectively). Thresholds were determined by visual inspection as the minimum sound or light intensity that elicited a reproducible response waveform in the recorded traces.

**Juxtacellular recordings from single putative SGNs.** For auditory nerve recordings[15], a glass microelectrode (~25 MΩ) was advanced through the posterior end of the anteroventral cochlear nucleus, aiming toward the internal auditory canal using an Inchworm micropositioner (EXFO Burleigh). Extracellular action potentials were amplified using an ELC-03XS amplifier (NPI Electronic, Tamm, Germany), filtered (band pass, 300–3000 Hz), and digitized (TDT System 3) using custom-written Matlab (Mathworks) software. Data were further analyzed and prepared for display off-line using custom-written Python (Python Software Foundation, Delaware, USA) and Matlab software. Once light-responsive fibers were encountered, stimulation was performed by means of 400 or 900 ms-long light-pulse trains at varying stimulation rates, leaving 100 ms inter-train recovery over 20 repetitions. Responses within the first 400 ms were then used for analyses. Only recordings for which the fibers generate at least five spikes per light-pulse train (on average across the 20 iterations recorded for each frequency tested on each fiber) were included. Phase-locking was quantified using the vector strength[33] and its significance tested with the Rayleigh test. If $L > 13.8$, the null hypothesis was rejected at the 0.001 significance level:[47] insignificant VS were set to 0. The spike jitter, defined as the standard deviation of spike latency in one period of stimulation, was calculated using a time window equal to the stimulation period. The hazard function of the temporal jitter was evaluated for each stimulation rate by simulating Poisson spike trains at discharge rates from 10 to 300 spikes/s. The spike probability is the ratio between the number of spikes and the number of light-pulses. The temporal jitter is the standard deviation of spike latency across trials.

**Data analysis.** The data were analyzed using Matlab (The MathWorks, Inc., Natick, MA, USA), Excel, Igor Pro 6 (Wavemetrics, Portland, OR, USA), Origin 9.0 (OriginLab, Inc., Northampton, MA, USA), and GraphPad Prism (GraphPad Software, La Jolla, CA, USA). Averages were expressed as mean ± s.e.m. or mean ± s.d., as specified. References to data in the main text were expressed as mean ± s.e.m. For statistical comparison between two groups, data sets were tested for normal distribution (the D'Agostino & Pearson omnibus normality test or the Shapiro–Wilk test) and equality of variances (F-test) followed by two-tailed unpaired Student's $t$-test, or the unpaired two-tailed Mann–Whitney U test when data were not normally distributed and/or variance was unequal between samples.

For evaluation of multiple groups, statistical significance was calculated by using one-way ANOVA test followed by Tukey's test for normally distributed data (equality of variances tested with the Brown–Forsythe test) or one-way Kruskal–Wallis test followed by Dunn's test for non-normally distributed data.

**Data availability.** The data that support the findings of this study and code used for analysis are available from the corresponding author upon reasonable request.

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

## Acknowledgements

We would like to thank Nicola Strenzke for co-supervising the work of David Lopez de la Morena. We would like to thank Ina Bartnik and Nicole Fürst for the preparation of the rat hippocampal neuron cultures. We also would like to thank Verena Pintschovius, Heike Fotis, Sandra Gerke and Christiane Senger-Freitag for excellent technical assistance, Helga Husmann for help with the preparation of the figures, Gerhard Hoch for programming stimulation software and Linda Hsu for preparing artwork (Fig. 4a). The Chrimson clone was kindly provided by Edward S. Boyden. This work was supported by the German Research Foundation Collaborative Research Centers 807 (to E.B.) and 889 (to To.M.)., Center of Excellence Frankfurt Macromolecular Complexes (to E.B.), the Center for Nanoscale Microscopy and Molecular Physiology of the Brain (to To.M.), by the Max Planck Society (to E.B. and J.J.L.), by the European Research Council (ERC) under the European Union´s Horizon 2020 research and innovation programme (grant agreement No 670759 - advanced grant "OptoHear" to To. M. - and grant agreement No 335587 - starting grant "AttentionCircuits" to J.J.L.)

## Author Contributions

T.Ma., P.G.W., J.J.L., T.Mo. and E.B designed research; T.Ma. (Fig. 1, Fig. 2, Table 1, Fig. S1-S6, Table S1-S4), D.L.M. (Figs. 4–6, Fig. S10-S11), V.S. (Fig. 3, Fig. S8, Fig. S9), J.S. (Table 1, Fig. S1, Table S1, Table S2), A.D. (Fig. 2, Fig. S7, Table S4), K.F. (Table S1), C.W. (Fig. 5), S.J. (Fig. 4), K.B (Figs. 4 and 5), VR (Fig. 4 and Fig. S10), LB (Fig. 4) and A.

H. (Fig. 6 and Fig. S11) performed research and analyzed data; J.J. prepared the AAVs; T. Ma., D.L.M., T.Mo. and E.B with contributions from V.S. and J.J.L. wrote paper.

## Additional information

**Competing interests:** E.B., T.Ma., T. Mo., P.G.W. and D.L.M. declare no competing non-financial interests but the following competing financial interests. E.B., T.Ma., T. Mo., P.G.W. and D.L.M. are authors on a pending world patent application related to this work, filed by Max-Planck-Gesellschaft zur Förderung der Wissenschaften E.V. and Universitaetsmedizin Goettingen (application no. PCT/EP2017/063458; priority date, June 3th 2016). E.B., P.G.W. and T.Ma. are authors on a pending world patent application related to this work, filed by Max-Planck-Gesellschaft zur Förderung der Wissenschaften E.V. (application no. PCT/EP2017/063425; priority date, June 3th2016). All other authors declare no competing interests.

