## [Peer Review File · Nature Communications]

Reviewers' comments:

Reviewer #1 (Remarks to the Author):

This study by a leading group in the channelrhodopsin field described 2 mutant variants of chrimson that has fast channel closure rate and retains the red-shifted action spectra. The v-chrimson has comparable properties to the original properties of chrimson and vf-chrimson have smaller photocurrent but faster kinetics. The authors demonstrated it is possible to trigger highly temporal precise spiking in cultured hippocampal neurons, parvalbumin-expressing interneurons in brain slices, spiral ganglia neurons in cochlea. Although I believe this manuscript is very exciting in the development of channelrhodopsin and this study potentially have high level of impacts in the field, the manuscript as it is presented at the current form have some major scientific faults that need to be addressed before it should be accepted for publication.

Major points:

1. Is the chrimson used in the study CsChrimson or Chrimson? Chrimson has a rather poor membrane trafficking and hence version with enhanced membrane trafficking (CsChrimson) was developed and used in the drosophila (developed by Vivek Jayaraman's group in Klapoetke et al Nat Methods 2014). With very little images for the cells expressing the eachannelrhodopsin, it is hard to be convinced of the statement 'high expression of the fast Chrimson mutants in neurons'. It would be useful to see the membrane expression of the tethered fluorescent protein in both cultured neurons and cell lines (and maximum projection should not be used).
2. The paper deals with very fast kinetic measurements with patch-clamping. This is not a trivial measurement so the recording needs to be done carefully and properly. It is difficult to judge from the information given whether this is done correctly. The only value given is the pipette resistance. The important factors that needs to be considered should be the access (series) resistance, the ratio with membrane resistance, the capacitance of the cell, whether series resistance compensation was used and what is the amount of compensation, what are the criteria for data elimination for the recording (e.g., when series resistance to membrane resistance ratio is below X etc.). This also has significant impacts on permeability ratio measurement as well. Without these information it is difficult to judge the quality of the experiments and measurements.
3. 'Chrimson mutants carrying the Y268F mutation showed reduced expression in NG cells' At the same time f-Chrimson was highly expressed in NG cells'. Data should be shown for this. The current densities should be adjusted to membrane fluorescence. The values in suppl table 2 is

very open to selection bias of the experimenter.

4. Voltage-clamp measurements in neurons has serious space-clamp issues and is generally a practise that should be avoided if possible. How are the space-clamp issues controlled or adjusted in Supplementary Table 4. Are these nucleated patches? Is the light limited to soma only? Are voltage-gated ion channels blocked (by a combination of cesium and other drugs)? Without even attempting to control the voltage-clamp, the off-rate time constant and current density values are questionable. In addition, the comments above regarding series resistance, capacitance and membrane resistance will also apply to the neurons and impacted it more significantly. What are these values and are the measurements justified?

5. For figure 2, how are the neurons selected for recording? Are these blindly selected to avoid bias towards expression levels? Are there cells that has expression levels that resulted in spiking failures and expression level too high for the cells to repolarize? It is surprising if all randomly selected neurons all express appropriate levels of protein to achieve such a high rate of firing.

6. For the parvalbumin neuron recording, 'irradiance... was adjusted individually for every neuron to cause reliable firing'. I understand the goal is to demonstrate it is possible to achieve reliable firing at high frequencies, however, I think this practise is scientifically questionable as this refers to a main figure and can be misleading.

7. The shape of the spikes for the SGN extracellular recording (figure 6b) is very unusual as extracellular recording would give a biphasic shape that is the derivative of the intracellular spike. The recording looks more like a partially impaled cell using sharp electrode. This raises issue with validity of the data and recording quality.

8. Supplementary figure 4 shows that the photocurrent from Chrimson and its derivatives suffer from inactivation/desensitization that never recovers. Given the current study utilizes repetitive stimulation that potentially would result in similar inactivation/desensitization, it is difficult to know how the stimulation that were used represented the initial peak current or not. Can the author shows the voltage clamp recordings of the cell to different frequencies of stimulation and the shape of the responses whether such inactivation/desensitization occur or not. This information would be crucial for potential users of the technology.

9. The kinetic measurement with a fast shutter still has a closing time of 0.7ms which can still be ~10% of the channel closing rate of f-Chrimson and vf-chrimson. Maybe a fast LED would be a better way to measure the rates. The authors would need to account for the limitations of the instrumentation on the measured values somehow if the shutters are used.

Minor points:

10. The X-axis of supplementary figure 6 should be in log scale. As this moment, the

compression of the axis makes it hard to see the differences.

11. Some language in the writing is rather awkward, some English editing would improve the readability of the manuscript. E.g., Helix F is moving during the transition from the closed to the open state of channelrhodopsin' => Closed to open state transition is associated with movement of helix F. 'the action spectra of f-chrimson and vf-chrimson were not shifted to lower wavelengths' => The action spectra of f-chrimson and vf-chrimson were not blue-shifted. 'short living' and 'long living' channels may need to be expanded as channel with short opening lifetime and long living lifetime as short living and long living are not scientific and could also refer to protein lifetime. 'extracellular recordings from single neurons – the gold standard of analysis'. The 'gold standard of analysis' can be removed.

12. ReaChR (ChR2CV1, CV2, Red absorbing Channelrhodopsin....) is actually ChR1/VChR1/VChR2 chimera and ReaChR stands for Red-activatable Channelrhodopsin.

13. The axis crossing in supplementary figure 2 should be at zero to make the value more readable.

Reviewer #2 (Remarks to the Author):

The manuscript by Mager et al. "High frequency neural spiking and auditory coding by ultrafast red-shifted optogenetics" from the labs of Ernst Bamberg and Tobias Moser represents an important step toward fast optogenetically driven neuronal activity and its use in sensory prosthesis, particularly cochlear implants. Although optical stimulation can be spatially highly specific - a major advantage compared to electrical stimulation which normally spreads uncontrollable - they normally lack temporal precision, which is absolutely pivotal for proper stimulation in many neuronal systems, notably in the auditory system. Moreover, phototoxicity also provides a challenge for the long-term use of channelrhodopsins. The new channelrhodopsins of the Chrimson variants presented in this manuscript are activated by red-light, which significantly reduces the problem of toxicity. Moreover, it is, as the manuscript shows by activation of various neuronal systems using different methods, comparably fast and therefore a potential candidate for cochlear implants. Some proof of concept for the auditory system is provided in the manuscript.

However, there are several major issues that need to be solved/addressed before the manuscript can be published. They mainly concern the way the data is presented, described and discussed, but also some missing statistical analyses.

Major concerns:

This is not the first time faster channelrhodopsins have been constructed and tested. While one of them, “Chornos” -type, are explicitly mentioned and discussed (although they were, as far as I know, never tested at similar high rates), ChETA (Gunaydin et al. 2010) is cited but not discussed (and is missing in Table 1) although it had been shown to activate neurons at least up to 200 Hz. This gives the overall impression that the actual use of Chrimson based on temporal precision is either far overstated or partially facilitated by hiding important information. This would leave the toxicity issue as the only justification for a higher impact publication.

The issue concerning the temporal precision is concerning for two other reasons: first, the crucial figure is hidden as supplement (Figure S10, Figure 6f is almost impossible to read), and hints for proper circular statistics (e.g. Rayleigh-Test) is missing (at least I could not find it). Moreover, Figure 6 and S10 leave us with questions: It is stated at several points throughout the manuscript that “high temporal precision”, nearly mimicked “sound-evoked SGN firing” (Discussion). Given that the mean vector strength of SGNs found for instance at 400 Hz optical stimulation rate was below 4 (Fig. S10), whereas auditory nerve fibers normally show a VS \gg 8 at 400 Hz acoustic stimulation (or even higher; compare Koepl 1997 J Neurosci), this appears as quite overstating - or did I miss something here? Moreover, Last paragraph, 4th sentence before last: it is referred to Supplement Figure 10. There it is stated in the figure legend that “first spike jitter” was analyzed. What does this mean? Only the first spike in a train was analyzed (which would be meaningless) or were there more spikes per photostimulation (which would be in strong contrast to the results in figure 3 and need an explanation). Besides this: the jitter is quite large: about one order of magnitude worse compared to 1 kHz auditory stimulation (see Koepl 1997 J Neurosci). Hence the statements about timing need to be more concise and transparent.

In general, in the Result part important numbers are frequently missing (the reader has to check the figures or even supplements) and the authors are rather generous with statements like “required more light compared to” (paragraph 5, 3rd sentence), “high expression of the fast Chrimson mutants” (same paragraph, three sentences down), or stating significance without giving the numbers in text. Worst case: “The temporal fidelity of high frequency neural photostimulation was considerably improved for the fast Chrimson mutants (Fig. 2a,b,d).” (same paragraph, last sentence). No specific numbers, and what means “considerably” without statistics?

The manuscript shows that different groups joined and put things together (which is great!), but that they did not put too much effort in making this a coherent manuscript. This is obvious in Abstract, Introduction and Discussion but most dramatically in the figures, which appear quite inhomogeneous. Some streamlining would be great.

Finally, there is a misuse of the term “coding” in the title and the text. Auditory “coding” has not been tested. “Signaling” has been tested, this would be the correct term (beside the more neutral

term “firing properties” as alternative). Hence, the title and MS at several points need to be changed accordingly.

As these comments seem harsh, it is all about wording/writing, after all. Fixing accordingly should improve the paper considerably.

Abstract:

The first sentence reads nice, but is rather commonplace.

5th sentence: “coding” needs to be replaced.

Introduction:

Para 2, 3rd sentence: not only speed matters, but also (or even more) precision!

Last sentence: what is an “excellent” expression?

Results:

See above – I am not listing every case of missing numbers or statistics.

Para 2, last sentence (“Structural information...”); discussion, not a result.

Para 7 (starting with “Using current injection..”), 3rd sentence: what exactly means “follows”

Second para after the headline “F-Chrimson is a promising candidate for hearing restoration”, sentences 4-8: this is too long an introduction for a Results sub-chapter

Last paragraph, 1st sentence: again misuse of “coding”.

Last paragraph, 4th sentence before last: it is referred to Supplement Figure 10. There it is stated in the figure legend that “first spike jitter” was analyzed. What does this mean? Only the first spike in a train (which would be meaningless) or were there more spikes per photostimulation (which would be in strong contrast to the results in figure 3). Besides this: the jitter is quite large: about one order of magnitude worse compared to 1 kHz auditory stimulation (see Koepl 1997 J Neurosci). Leaves me rather uneasy.....

Discussion:

Missing is a discussion of the lack of phase-locking in the mouse to pure tones and the nature of the response to trains of pulses or AM. What would that mean for human CIs?

Para1, last sentence: the discrepancy of figure 3 (non auditory) and supplement figure 10 needs to be resolved/explained before one can make such a statement.

Last para, 7th sentence (“In fact...”): I do not see the data from nerve fibers that qualify the statement of “high temporal precision, nearly mimicking sound-evoked SGN firing.”

Figure 1: scale bar for currents are missing

Figure 3: rather ample use of space – again. Abbr. “Hz” not “HZ”

Figure 4: here “d” and “e” could be enlarged, f reduced.

Figure 6: is rather busy. F needs to be enlarged, the circular panels are rather un-informative (given that phase-histograms are provided): instead significance values from a Rayleigh-Test should be provided..

Table 1: why is ChETA not listed – this would be at least as relevant as the others.

Reviewer #3 (Remarks to the Author):

I agree with the central pitch of this paper, that the very high speeds of driving spikes and precision in following that is achieved is an important new finding and should be published in a forum like Nature Communications. The paper at an abstract, logical level does a nice job of presenting the logic from basic biophysics up to systems implementation. What the paper does a borderline job of is presenting the origins of these findings in a clear way: It is often unclear how many recordings went into each trace or sigmoid, how the significance of differences was determined (e.g., between constructs) and even what data came from this paper or from other sources (?) Put simply, I am in favor of this work, but I have to understand much better the claims made, about facts stated in the text and about the data themselves.

I. A major shortcoming of the paper is the abrupt, almost casual way it discusses channel kinetics. This paper is most likely of greatest interest to systems neuroscientists that know enough biophysics to be able to follow the arguments, but not to fill in great unexplained blanks in the text. Specific examples where a few more sentences would help before abruptly diving into details are given, but there are many other places where the sudden introduction of terms and ideas without real transitions makes it very hard to read:

- a. Intro lines 63-68 or so: Unpacking the issues around fast light recruitment of power requirements would greatly help in understanding the challenge the paper is addressing. So, not a full explication, but at least a few more sentences.
- b. To simply presume that everyone knows what Helix F is completely unwarranted. So, explain a little bit at least about what it is at the outset of the Results. Again, just a few sentences would make the paper much more broadly impactful.
- c. SLOW DOWN! The entire paragraph from lines 106 to 117 is full abrupt statements that assume a great deal of very specific prior art about the channels, that likely does not exist much beyond 3-4 labs in the world. Explain why one mutation 'points' to another likely mechanism, how the calcium permeability was determined.
- d. Also, it is very unclear in this section which findings are truly new, and which are data incorporated from previous work. CatCh calcium permeability is, of course, well documented already—are they current observations a modification or extension of that prior work, or are the authors just narrating their internal logic?

II. The second major shortcoming is the way that the significance of data-based claims are made. For example, in Figure 2, they say that the sigmoids originate from "N = 7." They should in some way show us the variance across cells, not just the mean fit of the sigmoid. Similarly, error bars should be more systematically represented in Figure 5 (why are they only shown for some data points?—if the authors are worried about obstructing the data, then 1. Perhaps that is informative to know and 2. They can always plot them at lighter density, but we should be able to see them). Similarly, perhaps I missed it, but it is not at all clear on what basis the claims of Figure 5 are made—how many mice? What statistical tests were applied?

Responses to Reviewers' comments:

Reviewer #1 (Remarks to the Author):

This study by a leading group in the channelrhodopsin field described 2 mutant variants of chrimson that has fast channel closure rate and retains the red-shifted action spectra. The v-chrimson has comparable properties to the original properties of chrimson and vf-chrimson have smaller photocurrent but faster kinetics. The authors demonstrated it is possible to trigger highly temporal precise spiking in cultured hippocampal neurons, parvalbumin-expressing interneurons in brain slices, spiral ganglia neurons in cochlea. Although I believe this manuscript is very exciting in the development of channelrhodopsin and this study potentially have high level of impacts in the field, the manuscript as it is presented at the current form have some major scientific faults that need to be addressed before it should be accepted for publication.

We describe mutagenesis for faster ChRs, targeting four different ChRs, with the focus on the red absorbing Chrimson.

Major points:

1. Is the chrimson used in the study CsChrimson or Chrimson? Chrimson has a rather poor membrane trafficking and hence version with enhanced membrane trafficking (CsChrimson) was developed and used in the drosophila (developed by Vivek Jayaraman's group in Klapoetke et al Nat Methods 2014). With very little images for the cells expressing the eachannelrhodopsin, it is hard to be convinced of the statement 'high expression of the fast Chrimson mutants in neurons'. It would be useful to see the membrane expression of the tethered fluorescent protein in both cultured neurons and cell lines (and maximum projection should not be used).

The Chrimson used in the study is Chrimson not CsChrimson. The membrane-targeted expression of CsChrimson is not significantly bigger than the membrane targeted expression of Chrimson (Ref. Addendum and Corrigendum to Klapoetke, N. C. et al. Independent optical excitation of distinct neural populations. *Nat Methods* 11, 338-346, doi:10.1038/nmeth.2836 (2014)). As shown in Figure 1 of the Addendum the photocurrent of CsChrimson is not significantly larger than the photocurrent of Chrimson.

In order to determine the current density the photocurrent, which is measured at saturating intensities, is divided by the capacitance of the corresponding cell. The capacitance of the cell is proportional to the area of the cellular membrane and the photocurrent is proportional to the number of active channels in the cell membrane. Therefore, current densities, measured at saturating light intensities are a direct measure of membrane-targeted expression. Fluorescence values also report the fluorescence of inactive channels in intracellular compartments and, are, therefore a less specific measure of membrane-targeted expression. In response to the reviewer's comment we have appended confocal sections of f-Chrimson-YFP-expressing spiral ganglion neurons at high magnification demonstrating the abundance of the opsin in the spiral ganglion neuron membrane (Figure 4f).

2. The paper deals with very fast kinetic measurements with patch-clamping. This is not a trivial measurement so the recording needs to be done carefully and properly. It is difficult to judge from the information given whether this is done correctly. The only value given is the pipette resistance. The important factors that needs to be considered should be the access (series) resistance, the ratio with membrane resistance, the capacitance of the cell, whether series resistance compensation was used and what is the amount of compensation, what are the criteria for data elimination for the recording (e.g., when series resistance to membrane resistance ratio is below X etc.). This also has significant impacts on permeability ratio measurement as well. Without these information it is difficult to judge the quality of the experiments and measurements.

We have long lasting experience with electrophysiological experiments. All electrophysiological measurements were done carefully and properly. We are well aware that the crucial parameters mentioned from your side have to be controlled. In this regard improper measurements would lead to apparently slower off-kinetics and reduced photocurrents, which would underestimate the superior properties of our new mutants. We agree that the explicit mentioning of additional parameters is reasonable. Therefore a reference explaining the patch-clamp method (Line 410) and additional information regarding the procedure of electrophysiological measurements is added to the methods part (Lines 411-413, Lines 473-474)

3. 'Chrimson mutants carrying the Y268F mutation showed reduced expression in NG cells''At the same time f-Chrimson was highly expressed in NG cells'. Data should be shown for this. The current densities should be adjusted to membrane fluorescence. The values in suppl table 2 is very open to selection bias of the experimenter.

We are only interested in channel function. Fluorescence data can give only a guess about the expression level of the channel, but not about their function, which is directly obtained by electrophysiology: Photocurrent divided by single channel conductance x open probability. (Please also see answer B to question 1.)

4. Voltage-clamp measurements in neurons has serious space-clamp issues and is generally a practise that should be avoided if possible. How are the space-clamp issues controlled or adjusted in Supplementary Table 4. Are these nucleated patches? Is the light limited to soma only? Are voltage-gated ion channels blocked (by a combination of cesium and other drugs)? Without even attempting to control the voltage-clamp, the off-rate time constant and current density values are questionable. In addition, the comments above regarding series resistance, capacitance and membrane resistance will also apply to the neurons and impacted it more significantly. What are these values and are the measurements justified?

The measurements were performed in the whole-cell configuration (no nucleated patches). The light fiber illuminates an area of 0.125 mm² and was centered to the soma. Therefore, we can not fully exclude that peripheral parts of the axons are not illuminated in some cases. The voltage gated sodium channels were blocked by the addition of tetrodotoxin (See methods). The series resistance was <10 MΩ and the input resistance ranged from 0.7 GΩ to 3.5 G Ω (at V<20 mV). The mean capacitance of the measured cells was 35.4 ± 12.4 pF (n=31) (Line 474). Please also see answer to question 2.

5. For figure 2, how are the neurons selected for recording? Are these blindly selected to avoid bias towards expression levels? Are there cells that has expression levels that resulted in spiking failures and

expression level too high for the cells to repolarize? It is surprising if all randomly selected neurons all express appropriate levels of protein to achieve such a high rate of firing

For figure 2, neurons were blindly selected. Measurements from a specific, fast spiking neuronal subset, namely parvalbumin-positive interneurons, were performed in addition (Figure 3). Spike failures in the low frequency range were not an issue, which is probably due to good expression, fast kinetics and the application of short light pulses. As described in the text not all neurons achieve high rates of firing. (quote: 'The primary culture of rat hippocampal neurons comprises a multitude of different neuronal subtypes, most of which have a maximal firing frequency of 40-60 Hz⁹. Therefore, in most cases spike failures occurred at a frequency of 60 Hz (Fig. 2c). In single cases a frequency of 100 Hz was achieved (Fig. 2d).') (Lines 176-177)

6. For the parvalbumin neuron recording, 'irradiance... was adjusted individually for every neuron to cause reliable firing'. I understand the goal is to demonstrate it is possible to achieve reliable firing at high frequencies, however, I think this practise is scientifically questionable as this refers to a main figure and can be misleading.

The need to adjust irradiance or pulse duration individually is likely due to heterogeneous levels of expression achieved by AAV transduction, and potentially also due to different intrinsic properties of the interneurons and depth in the slice. These factors are not unique to our experiments, but rather a well known limitation of this form of optogenetics that has been covered in several papers (e.g. Klapoetke et al. (2014) Independent optical excitation of distinct neural populations. Nature methods; Gunaydin et al. (2010) Ultrafast optogenetic control. Nat Neurosci; Mattis et al. (2011) Principles for applying optogenetic tools derived from direct comparative analysis of microbial opsins. Nature methods). For instance, Klapoetke et al. (2014) report in their Fig. 4f that even for relatively controlled experiments in culture, the threshold irradiance for action potential initiation by 2 different channelrhodopsins varies by more than an order of magnitude between individual neurons, and this problem is exacerbated by the unprecedented range of stimulation frequencies in our current study. We therefore maintain that the presented data fully supports the conclusion that vf-Chrimson can elicit action potential firing up to the intrinsic frequency limit of parvalbumin-positive interneurons. However, we appreciate the reviewer's concern that this requirement needs to be more clearly pinpointed, and have added a sentence to that effect to the main text (line 193-194). In addition, we have prepared a new supplementary figure (9) that illustrates the need to adjust irradiance in order to achieve optimally precise activity control.

7. The shape of the spikes for the SGN extracellular recording (figure 6b) is very unusual as extracellular recording would give a biphasic shape that is the derivative of the intracellular spike. The recording looks more like a partially impaled cell using sharp electrode. This raises issue with validity of the data and recording quality.

Indeed, in our experience from *in vivo* recordings from single spiral ganglion neurons (SGN, e.g. example traces in Strenzke, N. et al. Complexin-I Is Required for High-Fidelity Transmission at the Endbulb of Held Auditory Synapse. J. Neurosci. 29, 7991-8004 (2009); Neef, J. et al. The Ca²⁺ Channel Subunit beta2 Regulates Ca²⁺ Channel Abundance and Function in Inner Hair Cells and Is Required for Hearing. J. Neurosci. 29, 10730 (2009); Buran, B. N. et al. Onset coding is degraded in auditory nerve fibers from mutant mice lacking synaptic ribbons. J. Neurosci. Off. J. Soc. Neurosci. 30, 7587-7597 (2010); Pangršič, T. et al. EF-hand protein Ca²⁺ buffers regulate Ca²⁺ influx and exocytosis in sensory hair cells.

Proc. Natl. Acad. Sci. 112, E1028-E1037 (2015)) we find waveforms that are mostly monophasic (or triphasic, with small negative deflections surrounding a large positive peak. In contrast, when we are on our path through the anteroventral cochlear nucleus (AVCN), we also hit principal AVCN neurons (bushy cells and multipolar cells: Chopper), which typically have biphasic spike shapes. This difference might be due to the different thickness of the myelin sheath. Indeed, we believe that our extremely thin micropipette tip would sometimes partially impale the thick myelin sheath of the auditory nerve fibers (central neurites of the SGNs), such that the recordings are "juxtacellular". Very rarely we, indeed, directly impale the auditory nerve fibers and then we observe a sharp decrease in the baseline (jump to between -30 and -70 mV), which is accompanied by a change in spike waveforms (much more irregular, DC offset between spikes). Such recordings, usually, are not stable, which likely also results from filling our electrode with 3M NaCl (detrimental for neuronal function in intracellular recordings). Therefore, we discard these rare and transient intracellular recordings. We switched from KCl to NaCl because we do not aim for intracellular recordings and we sometimes accidentally break our electrode by hitting the bone at the bottom of the electrode path and NaCl would be expected to cause less damage.

8. Supplementary figure 4 shows that the photocurrent from Chrimson and its derivatives suffer from inactivation/desensitization that never recovers. Given the current study utilizes repetitive stimulation that potentially would result in similar inactivation/desensitization, it is difficult to know how the stimulation that were used represented the initial peak current or not. Can the author shows the voltage clamp recordings of the cell to different frequencies of stimulation and the shape of the responses whether such inactivation/desensitization occur or not. This information would be crucial for potential users of the technology.

Desensitization is a general property of channelrhodopsins. A detailed investigation for the novel channelrhodopsin variants is far beyond the purpose of our manuscript (light dependence and frequency dependence for all the investigated variants). However for the high frequency response the desensitization corresponds to the inactivation at stationary light conditions, which is shown in supplementary figure 4, which we believe is sufficient information.

Desensitization appears in response to the first light pulses. These were generally test light pulses in vc or cc mode prior to the shown traces. The stimulation protocol was optimized regarding to pulse length and light intensity prior to the shown traces. The time between the light pulses and the waiting time between the different light pulse protocols is too short for the peak current to recover. Therefore the light stimulation is due to channel activity in the desensitized state (Please note, that the given current densities refer to the desensitized state). The extremely slow kinetics of the peak recovery is therefore a beneficial property regarding photocurrent stability. After the first light pulses the variability of the amplitude of the photocurrent is very low.

9. The kinetic measurement with a fast shutter still has a closing time of 0.7ms which can still be ~10% of the channel closing rate of f-Chrimson and vF-chrimson. Maybe a fast LED would be a better way to measure the rates. The authors would need to account for the limitations of the instrumentation on the measured values somehow if the shutters are used.

We agree, that a fast LED would be a better way to measure the rates. However the off-rates derived from shutter controlled measurement were not significantly different from the off-rates of photocurrents in response to ns-laser pulses. In the case where shutter closing affected the off-rate

(Chr88-K176R,S267M, Y261F at 34°C) ns-laser pulses were used. See methods part (Lines 437-441).

Minor points:

10. The X-axis of supplementary figure 6 should be in log scale. As this moment, the compression of the axis makes it hard to see the differences.

Supplementary figure 6 is modified accordingly.

11. Some language in the writing is rather awkward, some English editing would improve the readability of the manuscript. E.g., Helix F is moving during the transition from the closed to the open state of channelrhodopsin' => Closed to open state transition is associated with movement of helix F. 'the action spectra of f-chrimson and vf-chrimson were not shifted to lower wavelengths' => The action spectra of f-chrimson and vf-chrimson were not blue-shifted. 'short living' and 'long living' channels may need to be expanded as channel with short opening lifetime and long living lifetime as short living and long living are not scientific and could also refer to protein lifetime. 'extracellular recordings from single neurons - the gold standard of analysis'. The 'gold standard of analysis' can be removed.

The corresponding parts are changed according to your suggestions. (Lines 99-100, Lines 144-145, Lines 61-64, Line 261)

12. ReaChR (Chr2CV1, CV2, Red absorbing Channelrhodopsin...) is actually Chr1/VChR1/VChR2 chimera and ReaChR stands for Red-activatable Channelrhodopsin.

The corresponding part is corrected. (Line 369)

13. The axis crossing in supplementary figure 2 should be at zero to make the value more readable.

Supplementary Figure 2 is changed according your suggestion.

Reviewer #2 (Remarks to the Author):

The manuscript by Mager et al. "High frequency neural spiking and auditory coding by ultrafast red-shifted optogenetics" from the labs of Ernst Bamberg and Tobias Moser represents an important step toward fast optogenetically driven neuronal activity and its use in sensory prosthesis, particularly cochlear implants. Although optical stimulation can be spatially highly specific - a major advantage compared to electrical stimulation which normally spreads uncontrollable - they normally lack temporal precision, which is absolutely pivotal for proper stimulation in many neuronal systems, notably in the auditory system. Moreover, phototoxicity also provides a challenge for the long-term use of channelrhodopsins. The new channelrhodopsins of the Chrimson variants presented in this manuscript are activated by red-light, which significantly reduces the problem of toxicity. Moreover, it is, as the manuscript shows by activation of various neuronal systems using different methods, comparably fast and therefore a potential candidate for cochlear implants. Some proof of concept for the auditory system is provided in the manuscript.

However, there are several major issues that need to be solved/addressed before the manuscript can be published. They mainly concern the way the data is presented, described and discussed, but also some missing statistical analyses.

Major concerns:

This is not the first time faster channelrhodopsins have been constructed and tested. While one of them, "Chornos" -type, are explicitly mentioned and discussed (although they were, as far as I know, never tested at similar high rates), ChETA (Gunaydin et al. 2010) is cited but not discussed (and is missing in Table 1) although it had been shown to activate neurons at least up to 200 Hz. This gives the overall impression that the actual use of Chrimson based on temporal precision is either far overstated or partially facilitated by hiding important information. This would leave the toxicity issue as the only justification for a higher impact publication.

We did not mention, that the new Chrimsons are the first fast switching ChRs. This is confirmed by the second sentence of the reviewer! As quoted several times in the MS the focus is the fast red absorbing ChR with superior fast kinetics. However the reviewer is correct, that we have not explicitly mentioned ChETA in the main text. We now explicitly mention ChETA in the main text, but it is not inserted in the table. (Lines 85-87). This is not a review.

The issue concerning the temporal precision is concerning for two other reasons: first, the crucial figure is hidden as supplement (Figure S10, Figure 6f is almost impossible to read), and hints for proper circular statistics (e.g. Rayleigh-Test) is missing (at least I could not find it).

In response to the reviewer's concern we have performed further analysis of the recordings of single spiral ganglion neurons and have moved the spike jitter information to the main MS (now Figure 6e). This also includes the requested Rayleigh-test.

Moreover, Figure 6 and S10 leave us with questions: It is stated at several points throughout the manuscript that "high temporal precision", nearly mimicked "sound-evoked SGN firing" (Discussion). Given that the mean vector strength of SGNs found for instance at 400 Hz optical stimulation rate was below 4 (Fig. S10), whereas auditory nerve fibers normally show a VS $\gg 8$ at 400 Hz acoustic stimulation (or even higher; compare Koepl 1997 J Neurosci), this appears as quite overstating - or did I miss something here?

In our MS we followed the definition according to Goldberg and Brown, 1969 of vector strength: $r = 1/n \sqrt{[\sum \sin(a_i)]^2 + [\sum \cos(a_i)]^2}$

where a_i is the phase angle of spike i relative to the modulation cycle of the stimulus, and n is the total number of spikes in the analysis window (r can maximally reach 1). However, in physiology this is typically not reached even for low frequency tones due to jitter of firing of the neurons that is not (fully) locked to the acoustic stimulus. This is also the case in the study, cited by the reviewer, by Christine Köppl on the barn owl, which employs an auditory system with probably the highest temporal precision of signaling that outperforms all mammals studied (e.g. the famous Figure 11 of that study). In the present study we found a vector strength of 1 for most putative spiral ganglion neurons at a stimulation rate of 50 Hz and for several of them at 100 Hz. This high vector strength estimate likely resulted from i) these stimulus rates being lower than those typically tested acoustically (e.g. the overview figure 11 in Köppl, 1997 starts at around 200 Hz and also finds double spikes), ii) from the fact the putative spiral ganglion neurons studied here did not show spontaneous firing behavior (probably due to the cochlear dysfunction resulting from the surgery and insertion of the optical fiber into the scala tympani via the round window) and iii) from calculating the vector strength by relating the spike-timing to the period from the start of the

pulse to that of the next, which is not directly comparable with the period used during sound experiment. For clarity we have now included this discussion in the MS.

Vector strength declined at higher stimulation rates and similar to what is also found for acoustic stimulation of mouse spiral ganglion neurons. Given the high frequency cochlea of the mouse, this was performed by using transposed tones (amplitude-modulated tones) instead of low frequency tones. For the sake of the revised MS we decided to turn to published data which also covered 125 and 250 Hz but we removed the spike probability information as it was not part in the publication (Buran et al., 2010).

On request of the reviewer we have further specified the statement.

"Temporal precision of firing, evaluated based on vector strength (³³, see methods, Fig. 6c,d) and temporal jitter (i.e., standard deviation of spike latency across trials, Fig. 6e, Supplementary Fig. 11) varied between the recorded neurons and, generally, reflected a good precision. The vector strength declined with increasing stimulation rate up to 1 kHz. For a comparison, we re-plot the median vector strength of firing driven by amplitude-modulated tones in mouse SGNs (³⁴, Fig. 6d) used because phase-locking to pure tones is hard to achieve in the "high frequency" mouse cochlea³⁸. We note that the putative SGNs recorded in the present study typically did not fire spontaneously probably due to the ear surgery. Lack of spontaneous firing and the short (1 ms) and pulsatile optogenetic stimulation likely explain why vector strength tended to be higher for low stimulus rates when compared to amplitude-modulated tones, while it was comparable at 500 and 1000 Hz of pulse rate and modulation frequency, respectively. Temporal jitter was typically below a millisecond and tended to decline when increasing stimulus rates beyond 400 Hz (Fig. 6e). "

While it seems most appropriate to compare the temporal precision for optogenetic and acoustic SGN stimulation within the same species, we have also included reference to temporal precision of acoustic and electrical SGN stimulation in other species in the discussion section. Moreover, we toned down the statements on the temporal precision throughout the MS and provided reasoning on how it is limited by closing kinetics of f-Chrimson and the resulting relative refractoriness.

"The closing kinetics of f-Chrimson and the resulting relative refractoriness probably also limits the temporal precision of f-Chrimson-mediated SGN firing. The vector strength, a measure commonly used to analyze the extent of "phase-locking" in SGNs ³⁷, was comparable for mouse SGN firing with 500 and 1000 Hz of f-Chrimson-mediated optogenetic stimulation and with amplitude modulated tones³⁴. However, we note that vector strength and temporal jitter of f-Chrimson-mediated SGN firing in mice indicate a lower temporal precision than that of acoustic hearing and electric stimulation in species with prominent phase-locking of SGN firing^{32,37}.

Moreover, Last paragraph, 4th sentence before last: it is referred to Supplement Figure 10. There it is stated in the figure legend that "first spike jitter" was analyzed. What does this mean? Only the first spike in a train was analyzed (which would be meaningless) or were there more spikes per photostimulation (which would be in strong contrast to the results in figure 3 and need an explanation).

We apologize for the confusion caused by the chosen term "first spike latency" and "first spike latency jitter". We referred to the first spike per pulse in the time window until the next pulse commenced and to the standard deviation of spike latency across trials (temporal jitter). Indeed, as shown in the example neuron in Figure 6, we did rarely see more

than 1 spike per pulse. For clarity, we have now labeled the axis as "spike latency" and "temporal jitter" and have added this data to new Figure 6.

Besides this: the jitter is quite large: about one order of magnitude worse compared to 1 kHz auditory stimulation (see Köppl 1997 J Neurosci). Hence the statements about timing need to be more concise and transparent.

Köppl 1997 used temporal dispersion, calculated from the vector strength at a given tone frequency:

$$s = \frac{\sqrt{2(1-r)}}{2\pi} \frac{1}{f},$$

We prefer to report temporal jitter as the standard deviation of spike timing across trials, because the optical stimulus is pulsatile and therefore the definition of stimulus period not as straightforward as for tones. Nonetheless, if we do calculate it is comparable to our estimate of temporal jitter (see Figure of the letter below).

Figure: Comparison of temporal spike jitter and temporal dispersion of f-Chrimson mediated firing

Temporal dispersion of SGN firing declines with tone frequency and in the owl, the expert for temporally precise SGN coding, is 100 μ s or less for frequencies of 1 kHz and above, but greater for frequencies lower than 1 kHz (Figure 6A, Köppl 1997). As mentioned above the direct comparison of pulsatile optogenetic stimulation to stimulation with tones is a bit

difficult. If done, the temporal precision of f-Chrimson-mediated optogenetic stimulation of mouse SGNs is worse than that of nerve fibers in the owl by about a factor of two: we find that temporal jitter at 1 kHz stimulation with 0.5 ms stimuli is about twice (median: 210 μ s, mean: 210 \pm 70 μ s) the temporal dispersion in owl SGN for 1 kHz tones.

While it seems most appropriate to compare the temporal precision for optogenetic and acoustic SGN stimulation within the same species (mouse), we have also included reference to temporal precision of acoustic and electrical SGN stimulation in other species in the discussion section. Moreover, we toned down the statements on the temporal precision throughout the MS and provided reasoning on how it is limited by closing kinetics of f-Chrimson and the resulting relative refractoriness.

"The closing kinetics of f-Chrimson and the resulting relative refractoriness probably also limits the temporal precision of f-Chrimson-mediated SGN firing. The vector strength, a measure commonly used to analyze the extent of "phase-locking" in SGNs ³⁷, was comparable for mouse SGN firing with 500 and 1000 Hz of f-Chrimson-mediated optogenetic stimulation and with amplitude modulated tones³⁴. However, we note that vector strength and temporal jitter of f-Chrimson-mediated SGN firing in mice indicate a lower temporal precision than that of acoustic hearing and electric stimulation in species with prominent phase-locking of SGN firing^{32,37}."

In general, in the Result part important numbers are frequently missing (the reader has to check the figures or even supplements) and the authors are rather generous with statements like "required more light compared to" (paragraph 5, 3rd sentence), "high expression of the fast Chrimson mutants" (same paragraph, three sentences down), or stating significance without giving the numbers in text. Worst case: "The temporal fidelity of high frequency neural photostimulation was considerably improved for the fast Chrimson mutants (Fig. 2a,b,d)." (same paragraph, last sentence). No specific numbers, and what means "considerably" without statistics?

"required more light compared to"

The numbers are now given in the text. (Lines 163-164)

"high expression of the fast Chrimson mutants"

The current density value is given in the text. (Line 170)

"The temporal fidelity of high frequency neural photostimulation was considerably improved for the fast Chrimson mutants (Fig. 2a,b,d)."

The sentence is deleted. Proof for the statement is provided in discussion to Figure 3.

The manuscript shows that different groups joined and put things together (which is great!), but that they did not put too much effort in making this a coherent manuscript. This is obvious in Abstract, Introduction and Discussion but most dramatically in the figures, which appear quite inhomogeneous. Some streamlining would be great. Finally, there is a misuse of the term "coding" in the title and the text. Auditory "coding" has not been tested. "Signaling" has been tested, this would be the correct term (beside the more neutral term "firing properties" as alternative). Hence, the title and MS at several points need to be changed accordingly.

Done.

As these comments seem harsh, it is all about wording/writing, after all. Fixing accordingly should improve the paper considerably.

Abstract:

The first sentence reads nice, but is rather commonplace.

5th sentence: "coding" needs to be replaced.

Done.

Introduction:

Para 2, 3rd sentence: not only speed matters, but also (or even more) precision!

Last sentence: what is an "excellent" expression?

The last sentence is changed. We do not use the term "excellent" anymore.

Results:

See above - I am not listing every case of missing numbers or statistics.

Para 2, last sentence ("Structural information..."); discussion, not a result.

We decided to conflate our results at that point. Please note that the corresponding paragraph has been modified according to suggestions from reviewer 3.

Para 7 (starting with "Using current injection.."), 3rd sentence: what exactly means "follows"

We have replaced the term 'some cells followed' with 'some cells' action potentials followed'

Second para after the headline "F-Chrimson is a promising candidate for hearing restoration", sentences 4-8: this is too long an introduction for a Results sub-chapter

Done.

Last paragraph, 1st sentence: again misuse of "coding".

Done, we have shortened the sentence and omitted coding as requested.

Last paragraph, 4th sentence before last: it is referred to Supplement Figure 10. There it is stated in the figure legend that "first spike jitter" was analyzed. What does this mean? Only the first spike in a train (which would be meaningless) or were there more spikes per photostimulation (which would be in strong contrast to the results in figure 3).

We apologize for the confusion caused by the chosen term "first spike latency" and "first spike latency jitter". We referred to the first spike per pulse in the time window until the next pulse commenced and to the standard deviation of spike latency across trials (temporal jitter). Indeed, as shown in the example neuron in Figure 6, we did rarely see more than 1 spike per pulse. For clarity, we have now labeled the axis as "spike latency" and "temporal jitter" and have added this data to new Figure 6.

Besides this: the jitter is quite large: about one order of magnitude worse compared to 1 kHz auditory stimulation (see Koepl 1997 J Neurosci). Leaves me rather uneasy....

We prefer to report temporal jitter as the standard deviation of spike timing across trials, because the optical stimulus is pulsatile and therefore the definition of stimulus period not as straightforward as for tones. Nonetheless, if we do calculate it is comparable to our estimate of temporal jitter (Supplementary Figure 10). Temporal dispersion of SGN firing declines with tone frequency and in the owl, the expert for temporally precise SGN coding, is 100 μ s or less for frequencies of 1 kHz and above, but greater for frequencies lower than 1 kHz (Figure 6A, Köppl 1997). As mentioned above the direct comparison of pulsatile optogenetic stimulation to stimulation with tones is a bit difficult. If done, the temporal precision of f-Chrimson-mediated optogenetic stimulation of mouse SGNs is worse by about a factor of two: we find that temporal jitter at 1 kHz stimulation with 0.5 ms stimuli is about twice (median: 164 μ s, mean: 200 \pm μ s) the temporal dispersion in owl SGN for 1 kHz tones.

While it seems most appropriate to compare the temporal precision for optogenetic and acoustic SGN stimulation within the same species (mouse), we have also included reference to temporal precision of acoustic and electrical SGN stimulation in other species in the discussion section. Moreover, we toned down the statements on the temporal precision throughout the MS and provided reasoning on how it is limited by closing kinetics of f-Chrimson and the resulting relative refractoriness.

"The closing kinetics of f-Chrimson and the resulting relative refractoriness probably also limits the temporal precision of f-Chrimson-mediated SGN firing. The vector strength, a measure commonly used to analyze the extent of "phase-locking" in SGNs ³⁷, was comparable for mouse SGN firing with 500 and 1000 Hz of f-Chrimson-mediated optogenetic stimulation and with amplitude modulated tones³⁴. However, we note that vector strength and temporal jitter of f-Chrimson-mediated SGN firing in mice indicate a lower temporal precision than that of acoustic hearing and electric stimulation in species with prominent phase-locking of SGN firing^{32,37}."

Discussion:

Missing is a discussion of the lack of phase-locking in the mouse to pure tones and the nature of the response to trains of pulses or AM. What would that mean for human CIs?

In response to this and the previous comments of the reviewer we have rewritten this section of discussion (see previous point).

Para1, last sentence: the discrepancy of figure 3 (non auditory) and supplement figure 10 needs to be resolved/explained before one can make such a statement.

In response to this comment, we have decided to remove SGNs from that statement.

Last para, 7th sentence ("In fact..."): I do not see the data from nerve fibers that qualify the statement of "high temporal precision, nearly mimicking sound-evoked SGN firing."

This statement was replaced by the above quoted text and we have toned down the statements on temporal precision of f-Chrimson-mediated SGN firing throughout the MS.

Figure 1: scale bar for currents are missing

Figure 1 shows normalized currents for better comparison.

Figure 3: rather ample use of space - again. Abbr. "Hz" not "HZ"

Done.

Figure 3 is changed accordingly.

Figure 4: here "d" and "e" could be enlarged, f reduced.

Done.

Figure 6: is rather busy. F needs to be enlarged, the circular panels are rather un-informative (given that phase-histograms are provided): instead significance values from a Rayleigh-Test should be provided..

Done.

Table 1: why is ChETA not listed - this would be at least as relevant as the others.

ChETA is not listed, because the table shows the new mutants we have developed and a mutant which is relevant for the description of the influence of the probable interaction between helix C and helix F on the closing kinetics of channelrhodopsins (namely Chr2 L132C). Table 1 is not reviewing prior existing channelrhodopsin variants. We included ChETA into the introduction.

Reviewer #3 (Remarks to the Author):

I agree with the central pitch of this paper, that the very high speeds of driving spikes and precision in following that is achieved is an important new finding and should be published in a forum like Nature Communications. The paper at an abstract, logical level does a nice job of presenting the logic from basic biophysics up to systems implementation. What the paper does a borderline job of is presenting the origins of these findings in a clear way: It is often unclear how many recordings went into each trace or sigmoid, how the significance of differences was determined (e.g., between constructs) and even what data came from this paper or from other sources (?) Put simply, I am in favor of this work, but I have to understand much better the claims made, about facts stated in the text and about the data themselves.

I. A major shortcoming of the paper is the abrupt, almost casual way it discusses channel kinetics. This paper is most likely of greatest interest to systems neuroscientists that know enough biophysics to be able to follow the arguments, but not to fill in great unexplained blanks in the text. Specific examples where a few more sentences would help before abruptly diving into details are given, but there are many

other places where the sudden introduction of terms and ideas without real transitions makes it very hard to read:

a. Intro lines 63-68 or so: Unpacking the issues around fast light recruitment of power requirements would greatly help in understanding the challenge the paper is addressing. So, not a full explication, but at least a few more sentences.

We now address the issues around fast light recruitment of power requirements in greater detail. (Lines 61-69)

b. To simply presume that everyone knows what Helix F is completely unwarranted. So, explain a little bit at least about what it is at the outset of the Results. Again, just a few sentences would make the paper much more broadly impactful.

The probable role of helix F in microbial type rhodopsins is explained in more detail at the outset of the results. (Lines 99-102)

c. SLOW DOWN! The entire paragraph from lines 106 to 117 is full abrupt statements that assume a great deal of very specific prior art about the channels, that likely does not exist much beyond 3-4 labs in the world. Explain why one mutation 'points' to another likely mechanism, how the calcium permeability was determined.

Additional sentences are added, which explain how calcium permeability was determined and which compare the results of the mutations in order to draw a clearer picture. (Lines 113-114) (Lines 118-119)

d. Also, it is very unclear in this section which findings are truly new, and which are data incorporated from previous work. CatCh calcium permeability is, of course, well documented already—are they current observations a modification or extension of that prior work, or are the authors just narrating their internal logic?

All findings in this section but the reference to CatCh are new.

II. The second major shortcoming is the way that the significance of data-based claims are made. For example, in Figure 2, they say that the sigmoids originate from "N = 7." They should in some way show us the variance across cells, not just the mean fit of the sigmoid.

The sigmoids in Figure 2 originate from 7 different cells in order to show the variability of neural photostimulation. It does not show a mean fit. We now explicitly mention that in the figure caption.

Similarly, error bars should be more systematically represented in Figure 5 (why are they only shown for some data points?—if the authors are worried about obstructing the data, then 1. Perhaps that is informative to know and 2. They can always plot them at lighter density, but we should be able to see them). Similarly, perhaps I missed it, but it is not at all clear on what basis the claims of Figure 5 are made—how many mice? What statistical tests were applied?

The individual mice are shown in grey color (n=5) in panels showing analysis of oABR (Fig. 5d-i). Error bars are presented for averaged data (orange lines) only. Statistical tests and number of mice are included in the text now.

Reviewers' comments:

Reviewer #1 (Remarks to the Author):

Although some points raised are now addressed, many have not been adequately addressed yet. The authors would need to redo/check the line numbers their rebuttal refer to before resubmission. This makes the assessment really time-consuming and difficult. I also have trouble understanding some statements made either in the rebuttal or in the manuscript and would like to see more clarification.

Major points:

1 & 3. This is still not adequately addressed. Unfortunately, Addendum is not usually peer-reviewed as the main article so the quality of the comparison needs to be reproduced. The Addendum also has the statement 'However, we observed more cytosolic aggregates with the KGC version and a reduction of aggregates with the ER2 version' so this is not really a settled issue. In addition, the measurements of photocurrent in neurons in the Klapoetke et al. paper are problematic given the experimenter's bias (point 5) and space clamp issue (point 4). Can the authors show some images (randomly selected) in cell lines and cultured neurons of the expression pattern (cultured neurons are preferred over in vivo due to the higher resolution and quality)? Are the authors not showing this because the surface expression is poor? It is correctly pointed out that fluorescence measurements have their own issues, however, the key of showing typical images is to know how the comparison to other variants and photocurrents are done and how the experimenter's bias are avoided when there are various degree of membrane trafficking and expression levels. Regarding point 3, any statements regarding the surface expression should be backed up by experimental results. If the photocurrents are not adjusted to membrane fluorescence, are the cells selected for recordings randomly selected without using the fluorescence to make sure there is no experimenter's bias in cell selection.

2. Can the author please check the lines they refer to. Can the same parameter for NG108-15 cells be listed in the method section as well (membrane resistance, access resistance, capacitance etc.). What is meant under Electrophysiology recordings from cultured hippocampal neurons by the statement (at $V < 20\text{mV}$) (line 484)?

3. I have preferred to see the voltage-clamp recording in neurons done in solution that blocked potassium currents (e.g., Cesium-based solution with TEA) in addition to TTX and the illumination limited to soma only. Alternatively, the author should point out that space-clamp issues in neurons were not controlled in the main text, especially that a big range of capacitance and membrane resistance are observed. Are the capacitance and membrane resistance of the different groups comparable? If not, this can severely alter the interpretations of the results.

4. I have trouble understanding this statement 'Measurements from a specific, fast spiking

neuronal subset, namely parvalbumin-positive interneurons, were performed in addition (Figure 3)'. Can the authors clarify this statement. Does this contradict with their statement 'neurons were blindly selected'. Do the authors mean they were able to select parvalbumin interneurons by their morphology under bright field without the use of fluorescence (so they can select them blindly)?

5. I have expressed that I am not comfortable with this section due to how the experiments were done. A stronger statement should be included in the main text and figure legend such as 'if irradiation intensity is carefully adjusted for individual cells to their relative expression level, it is possible to achieve high fidelity.....' Currently it still seem a bit misleading.

7. Minor point: Isn't the typical extracellular recording of spike use negative voltage for the spike?

8. This information in the rebuttal may/should be in the main text.

Reviewer #2 (Remarks to the Author):

Overall the authors have done a very good job in addressing the reviewers' concerns. I have only two issues left.

1. Concerning my first major concern: I appreciate that ChETA is now mentioned in the text (although rather en passant) but I do not understand why it is not listed in the table. This is not about reviewing everything known, but rather how open one is in dealing with what is already out in the literature. Therefore, I do not understand why it is a problem putting it into the table.

2. The response to my concern related to the vector strength shown in figure 6 (and supplement) is a little strange. First, this reviewer knows circular statistics including the Goldberg and Brown analysis, lecturing here is unnecessary. Second, the response shows some unawareness concerning the problems in using circular statistics for measuring temporal precision of neuronal responses. It does, for instance, matter whether there are one or more spikes per cycle, which is particularly the case in responses to SAM. This causes systematic underestimations of response to SAM which can be compensated by either eliminating all but the first spike per cycle or by (better) using the jitter of the first spike only. Since the authors now provide the jitter, I am happy with the data and how it is presented. The discussion, however, could be improved in this regard.

I trust the authors will deal with this appropriately and I therefore do not need to see any changes except in the final publication.

Reviewer #4 (Remarks to the Author):

The authors have done a good and thorough job responding to the previous reviewer critiques.

Responses to Reviewers:

Reviewer #1 (Remarks to the Author):

Although some points raised are now addressed, many have not been adequately addressed yet. The authors would need to redo/check the line numbers their rebuttal refer to before resubmission. This makes the assessment really time-consuming and difficult. I also have trouble understanding some statements made either in the rebuttal or in the manuscript and would like to see more clarification.

Major points:

1 & 3. This is still not adequately addressed. Unfortunately, Addendum is not usually peer-reviewed as the main article so the quality of the comparison needs to be reproduced. The Addendum also has the statement 'However, we observed more cytosolic aggregates with the KGC version and a reduction of aggregates with the ER2 version' so this is not really a settled issue. In addition, the measurements of photocurrent in neurons in the Klapoetke et al. paper are problematic given the experimenter's bias (point 5) and space clamp issue (point 4). Can the authors show some images (randomly selected) in cell lines and cultured neurons of the expression pattern (cultured neurons are preferred over in vivo due to the higher resolution and quality)? Are the authors not showing this because the surface expression is poor? It is correctly pointed out that fluorescence measurements have their own issues, however, the key of showing typical images is to know how the comparison to other variants and photocurrents are done and how the experimenter's bias are avoided when there are various degree of membrane trafficking and expression levels. Regarding point 3, any statements regarding the surface expression should be backed up by experimental results. If the photocurrents are not adjusted to membrane fluorescence, are the cells

selected for recordings randomly selected without using the fluorescence to make sure there is no experimenter's bias in cell selection.

The reviewer's comment that the measurements from the Boyden lab, which were published in the Klapoetke et al., paper are biased, seems irrelevant to our work. Surface expression of the novel Chrimson variants is not poor, which is proven by the determined current densities (Supplementary Table 2 and 4). Current densities, measured at saturating light intensities are a direct measure of membrane-targeted expression. Fluorescence values, instead, also report the fluorescence of inactive channels in intracellular compartments (at least when not using super-resolution imaging) and, are, therefore a less specific measure of membrane-targeted expression. We already clarified in our previous response saying that "neurons were blindly selected" for patching in primary neuronal cultures, in order to avoid an experimental bias in cell selection. That non-biased way of cell choice likely causes the large standard deviations of the current density values given in Supplementary Tables 2 and 4 and the large variability of the dependence of the spike probability on the light intensity shown in Figure 2 e-g. The reviewer is right that we should explicitly mention the non-biased way of cell choice in our manuscript. Therefore we added the sentence "In order to avoid an experimental bias, the neurons for the electrophysiological recordings were chosen independent of the brightness of their EYFP fluorescence." to the methods part.

For the in vivo experiments, for which we don't provide current density values, we provide confocal images in the following. They show clear plasma membrane expression of f-Chrimson in spiral ganglion neurons following AAV-mediated gene transfer in vivo. Importantly, we did not find cell loss or decline of f-Chrimson expression in 9 months old mice (following early postnatal AAV-injection), which argues for good cell viability and reliability of the single short optogenetic manipulation.

Figure: Confocal Z-sections (0.25 μ m step) showing YFP expression (green) in parvalbumin positive SGNs (magenta) from cochlear cryosections on the AAV2/6-injected ear of a 9-months old C57Bl6/J mouse.

2. Can the author please check the lines they refer to. Can the same parameter for NG108-15 cells be listed in the method section as well (membrane resistance, access resistance, capacitance etc.). What is meant under Electrophysiology recordings from cultured hippocampal neurons by the statement (at $V < 20$ mV) (line 484)?

We now give the corresponding parameters for NG108-15 cells in the methods section. The term at $V < 20$ mV was erroneously added and is removed. The membrane resistance was determined from the dark current of current density measurements in hippocampal cells which were performed at $V = -70$ mV (Methods and Supplementary Table 4).

3. I have preferred to see the voltage-clamp recording in neurons done in solution that blocked potassium currents (e.g., Cesium-based solution with TEA) in addition to TTX and the illumination limited to soma only. Alternatively, the author should point out that space-clamp issues in neurons were not controlled in the main text, especially that a big range of capacitance and membrane resistance are observed. Are the capacitance and membrane resistance of the different groups comparable? If not, this can severely alter the interpretations of the results.

The big range of capacitance and membrane resistance values is due to the heterogeneity of the primary neuronal culture. We already wrote in the main text: “The investigation of neural photostimulation in the high frequency range is impeded by the heterogeneity of the primary neuronal culture. Therefore, we conducted patch-clamp experiments on parvalbumin-positive

interneurons heterologously expressing vf-Chrimson.” A grouping of neuronal subtypes in the primary culture based on the capacitance and the membrane resistance is far beyond the purpose of our manuscript. In response to the reviewer’s comment we added the sentence “Please note, that potential space clamp problems might have lowered the determined current density values. “

4. I have trouble understanding this statement ‘Measurements from a specific, fast spiking neuronal subset, namely parvalbumin-positive interneurons, were performed in addition (Figure 3)’. Can the authors clarify this statement. Does this contradict with their statement ‘neurons were blindly selected’. Do the authors mean they were able to select parvalbumin interneurons by their morphology under bright field without the use of fluorescence (so they can select them blindly)?

Neurons were blindly chosen in primary neuronal cultures. Parvalbumin positive interneurons were identified by their tdTomato fluorescence. We wrote in the main text: “The investigation of neural photostimulation in the high frequency range is impeded by the heterogeneity of the primary neuronal culture. Therefore, we conducted patch-clamp experiments on parvalbumin-positive interneurons heterologously expressing vf-Chrimson. Parvalbumin-positive interneurons display a fast spiking phenotype, and predominantly supply inhibition to the perisomatic domain of other neurons¹⁴. Heterologous expression of vf-Chrimson was achieved by intracerebroventricular injection of AAVs in transgenic mice that expressed tdTomato under the control of the parvalbumin promotor. Therefore, parvalbumin-positive interneurons could be identified in neocortical brain slices by their red fluorescence (**Supplementary Fig. 8a**).” Parvalbumin-positive interneurons were chosen blindly with respect to the vf-Chrimson expression.

5. I have expressed that I am not comfortable with this section due to how the experiments were done. A stronger statement should be included in the main text and figure legend such as ‘if irradiation intensity is carefully adjusted for individual cells to their relative expression level, it is possible to achieve high fidelity.....’ Currently it still seem a bit misleading.

We continue to agree with the reviewer that the heterogeneous expression levels achieved by AAV transduction, along with different depth of the neurons in the slice and hence different local light intensities during single photon stimulation make it necessary to adjust irradiance individually for each recording if the aim is to quantify the *achievable stimulation fidelity*, as has been shown in many previous studies (e.g. Klapoetke et al., 2014; Gunaydin et al., 2010; Mattis et al., 2011). In addition to the changes we have made to the last submitted version in response to the reviewer’s comments, and the new supplementary figure we have included on that issue, in the current version we have now added the following statement to the main text and figure legend as suggested by the reviewer: ‘We note that similar to previous work using AAV transduction and single photon stimulation⁶, it was necessary to adjust irradiation intensity individually for each neuron to achieve optimal stimulation fidelity (**Supplementary Fig. 9**).’

7. Minor point: Isn’t the typical extracellular recording of spike use negative voltage for the spike?

We assume that most of our recordings are juxtacellular, which are expected to give rise to positively polarized spikes (Gold et al. J Neurophys, 2009). In response to the reviewer's comment, we have performed an extensive review of spikes of auditory nerve fibers (ANF) from our own and work published by others (which not always contains original traces). We have double-checked the set-up to rule out unintended sign-inversion, however this is not the case and the same waveforms were also seen by in other labs including one of the co-authors at a completely different set-up in Montpellier. In summary, all the work involving sharp microelectrodes or loose patch-clamp we reviewed, reports sharp rising positive peaks. We would like to start with a statement of Ian Winter and Allan Palmer in their paper: Winter and Palmer, Hearing research (1990) on how they distinguished spikes of cochlear nucleus neurons from those of ANF: "Waveforms of action potentials from fibres (*ANF this is*) were monophasic positive going, invariably characterised by short rise times, crisp sound and were lost with small movements of the hydraulic microdrive."

Next, we present data from our work:

Mouse ANF, same recording conditions as in present study:

Fig. 4E from Strenzke et al. (2009) $Ca_v1.3$ channels in Complexin-I is required for high-fidelity transmission at the endbulb of Held auditory synapse J Neurosci 33 29(25):7991-8004.

Representative auditory nerve fiber recordings from CPX $I^{+/+}$ (black) and CPX $I^{-/-}$ (red) mice illustrating sound-evoked (50 ms tone burst at CF; 30 dB re threshold; 200 repetitions) and spontaneous (50 ms silence) spikes with good (left) and acceptable (right) amplitudes (5 repetitions each). Recordings with inferior signal-to-noise ratios were excluded.

Acoustically elicited spikes of ANF recorded on a WT C57Bl6/J mouse by David Lopez de la Morena

Acoustically elicited spikes of ANF recorded from a WT C57Bl6/J mouse by Dr. Tanvi Butola

Acoustically elicited ANF spikes recorded on a gerbil by Dr. Antoine Huet in Montpellier

Gerbil auditory nerve single unit traces, filter [0.1 10] kHz
Institut des Neurosciences de Montpellier, INSERM U1051

Acoustically elicited ANF spikes recorded on a gerbil by Dr. Antoine Huet in Göttingen

Gerbil auditory nerve single unit traces, unfiltered
Institut for Auditory Neuroscience, Göttingen

Acoustically elicited spikes of anteroventral cochlear nucleus (AVCN) recorded from a WT C57Bl6/J mouse by Dr. Tanvi Butola

Loose-patch recordings from spiral ganglion neurons in the explanted rat cochlea:

Fig. 1 from Wu JS et al. (2016) Maturation of spontaneous firing properties after hearing onset in rat auditory nerve fibers: spontaneous rates, refractoriness, and interfiber correlations. *J Neurosci* 36:10584–10597. 10.1523/JNEUROSCI.1187-16.2016

We assume that most of our recordings are juxtacellular, reporting monophasic positive going spikes of substantial amplitudes (up to few mV). Monophasic positive going spikes from juxtacellular recordings have been reported in the literature (see Fig. 4 Ebbesen et al. *Cell Reports*, 2016) and they are normally found in the auditory nerve (see above).

“Pure” extracellular recordings should in most cases report lower amplitude (Gold et al J Neurophys 2009). It is important to mention that most of the published extracellular example traces were recorded in CNS, with spikes arising from big neurons with thick axons, from which action potentials can be distinguished from baseline noise with lower impedance electrodes (1-10 MΩ) at a greater distance than seems feasible with ANFs.

Very rarely we, indeed, directly impale the auditory nerve fibers and then we observe a sharp decrease in the baseline (jump to between -30 and -70 mV), which is accompanied by a change in spike waveforms (much more irregular, DC offset between spikes). Such recordings, usually, are not stable, which likely also results from filling our electrode with 3M NaCl (detrimental for neuronal function in intracellular recordings). Therefore, we discard these rare and transient intracellular recordings. We switched from KCl to NaCl because we do not aim for intracellular recordings and we sometimes accidentally break our electrode by hitting the bone at the bottom of the electrode path and NaCl would be expected to cause less damage.

In summary, we hope that this data review makes the point that we the optogenetically driven spikes are similar to those we and others record with acoustic stimulation or as spontaneous activity. We have now modified the MS to state the juxtacellular recording configuration.

8. This information in the rebuttal may/should be in the main text.

Reviewer #2 (Remarks to the Author):

Overall the authors have done a very good job in addressing the reviewers' concerns. I have only two issues left.

1. Concerning my first major concern: I appreciate that ChETA is now mentioned in the text (although rather en passant) but I do not understand why it is not listed in the table. This is not about reviewing everything known, but rather how open one is in dealing with what is already out in the literature. Therefore, I do not understand why it is a problem putting it into the table.

The table shows the new mutants we have developed and the corresponding wildtypes. We show a mutant, which is relevant for the description of the influence of the probable interaction between helix C and helix F on the closing kinetics of channelrhodopsins in addition, namely ChR2 L132C.

2. The response to my concern related to the vector strength shown in figure 6 (and supplement) is a little strange. First, this reviewer knows circular statistics including the Goldberg and Brown analysis, lecturing here is unnecessary. Second, the response shows some unawareness concerning the problems in using circular statistics for measuring temporal precision of neuronal responses. It does, for instance, matter whether there are one or more spikes per cycle, which is particularly the case in responses to SAM. This causes systematic underestimations of response to SAM which can be compensated by either eliminating all but the first spike per cycle or by (better) using the jitter of the first spike only. Since the authors now provide the jitter, I am happy with the data and how it is presented. The discussion, however, could be improved in this regard.

I trust the authors will deal with this appropriately and I therefore do not need to see any changes except in the final publication.

In response to the reviewers comment we have move the interpretation of the comparison to the discussion section and extended to account for the comment. It now reads:

“The vector strength, a measure commonly used to analyze the extent of “phase-locking” in SGNs³⁸, was comparable for mouse SGN firing with 500 and 1000 Hz of f-Chrimson-mediated optogenetic stimulation and with transposed tones³⁴. We note that the SGNs recorded in the present study typically did not fire spontaneously probably due to the ear surgery. Lack of spontaneous firing the short (1 ms) and pulsatile optogenetic stimulation typically evoking a single spike likely explain why vector strength tended to be higher for low stimulus rates when compared to transposed tones, for which several spikes were generated per stimulus cycle. Moreover, vector strength and temporal jitter of f-Chrimson-mediated SGN firing in mice indicate a lower temporal precision than that of acoustic hearing and electric stimulation in species with prominent phase-locking of SGN firing^{32,38}.”

Reviewer #4 (Remarks to the Author):

The authors have done a good and thorough job responding to the previous reviewer critiques.

Reviewers' comments:

Reviewer #1 (Remarks to the Author):

From the author's response, is it correct that 'blind' selection of cell for recording is only done in neurons but not in NG108-15 cells. The authors would need to state this in their comparison of current density of different variants (Supp table 2) that the current density measurement is subject to experimenter's selection bias as this is not corrected for expression level on the membrane and cells are selected based on observable membrane fluorescence.

Can the authors check their reference on GHK equation for calculating calcium permeability. The original GHK equation does not take account of calcium due to its divalent nature which complicated the calculation. Can the authors also elaborate within their manuscript regarding the significant calcium permeability they measured which is contradicting a very recent publication by their rival Peter Hegemann's group (Vierock et al. Sci Report 2017) which detected no/minimal calcium permeability with Chrimson. Given how different they are, this result should be commented/explained.

The neuronal traces in Supplementary Figure 9 is a bit odd in the amplitudes of action potential varies significantly (the middle trace shows ~100mV as expected and top and bottom show ~80mV). The thresholds for action potentials (or the resting membrane potentials) also varies significant between the middle and the other two traces. Similarly in Figure 3 of the main text, the firing threshold of the parvalbumin cell in panel (a) appears to be much higher than the parvalbumin cell(s) shown in (c) (40 mV vs 10mV above resting membrane potential), is this correct or is the membrane potential very different in these cases? Can they provide evidences these are presented or measured correctly.

I believe the authors have misinterpreted the comments regarding membrane resistance and capacitance. As the authors are comparing the performance of the chrimson mutants in Fig 2 e-g and Suppl table 4 with very low number of cells (n=7), it is important to show that the membrane resistance and capacitance of these 3 groups are comparable to make sure the comparison is justified (scatter plot would be preferred over bar graphs or mean +/- SD). It would be even better if they can show that expression levels are comparable but this is harder to quantify accurately so it is not as crucial. Alternatively, the authors can randomly record from higher number of cells (~15-20 in each group) to justify the analysis statistically.

The images provided in the rebuttal on the membrane expression appears to be much better (in resolution, size and clarity) than the ones provided in the insets in the main figure or the current supplemental figure. These should go into the supplemental material.

Minor points on presentation

It is not totally correct that space clamp issue lead to underestimation of the current density. Space clamp issue leads to lack of clamping in distal dendrites (and axon) that can either lead to over or underestimation of the current density depending on what happens at distal processes.

The use of J to represent two different values (current density and illumination power) in supplemental table 4 is very confusing.

What cells are used in generating Suppl fig 6. Is this done in neurons since it is under the paragraph on neurons?

I should remind the authors to do a better job of editing their manuscript as stated previously. The followings are just some examples of the current version and not the complete list that needs to be addressed. In some lesser cases, they are very awkward to read, in more severe cases they are scientifically incorrect.

Can authors clarify this very long running sentence (Ln 523) ‘Data were acquired with a Multiclamp 700B amplifier in loose seal cell-attached (n=4) or whole-cell current-clamp mode (n=3 included into light-response data + n = 5 additionally included to the input-output curves in Fig. 3B). This is very confusing statement that needs to be rewritten in a clear manner.

Ln 43 ‘We developed structure-based mutants leading to a unifying concept’ The phasing is incorrect here. I believe they mean ‘We developed a homologous structure model that allow us to develop mutants’. ‘Structure-based mutants leading to a unifying concept’ as a sentence makes little sense scientifically.

Ln 47 ‘Because red light causes low light scattering....’. The correct phasing should be ‘Because red light has lower light scattering’. Light scattering is the result of the non-homogenous and non-transparent nature of tissue (so it is caused by tissue) and not because light itself ‘causes’ scattering.

Ln 51 ‘They drive spiking.....’ What is meant by ‘they’, does this refer optical cochlear implants or Chrimson or the system the authors developed (both the implants and chrimson variant). I suspect it is the later. Please rephrase the sentence to make this understandable

Ln 63 and 67 ‘more light’. I think it is more accurate to either refer to stronger light or more photons instead of the ambiguous ‘more light’.

Ln 113 'Permeability ratios were calculated according to the Goldman-Hodgkin-Katz equation by the shift of the reversal potential due to the exchange of external sodium by calcium.' This is another long running sentence with incorrect phasing. The following suggestion can be considered (although the authors should further work on it. 'Permeability ratios were calculated using the Goldman-Hodgkin-Katz equation with the measured values of the reversal potentials after replacing external sodium by calcium.'

Ln 115 '...calcium permeability could be verified in....' maybe just use 'is also verified'.

Ln 138 'carries the K176R mutation in addition' => 'carries the additional K176R mutation'

Ln 159, Ln 169. The use of 'proved' and 'proven' should be avoided in scientific writing. Scientifically proven something would require a higher level of testing than currently presented.

Can the authors clarify the sentence 'Optical stimulation was performed by a LED coupled to the objective (coolLed)' In 528. I assume they mean the LED was coupled to the microscope into the optical light path instead of a direct coupling of LED to the objective to excite the specimen.

Responses to Reviewers:

Reviewers' comments:

Reviewer #1 (Remarks to the Author):

From the author's response, is it correct that 'blind' selection of cell for recording is only done in neurons but not in NG108-15 cells. The authors would need to state this in their comparison of current density of different variants (Supp table 2) that the current density measurement is subject to experimenter's selection bias as this is not corrected for expression level on the membrane and cells are selected based on observable membrane fluorescence.

The reviewer is correct that we have omitted a detailed description of how the NG cells were selected for recording, and how their current density was determined. This has now been added to the section "Electrophysiological recording on NG108-15 cells" as follows: "The current density ($J_{-60\text{ mV}}$) was determined by dividing the stationary current in response to a 500 ms light pulse with a saturating intensity of 23 mW/mm² by the capacitance of the cell. In order to avoid an experimental bias, the NG108-15 cells for the electrophysiological recordings were chosen independent of the brightness of their EYFP fluorescence."

Can the authors check their reference on GHK equation for calculating calcium permeability. The original GHK equation does not take account of calcium due to its divalent nature which complicated the calculation. Can the authors also elaborate within their manuscript regarding the significant calcium permeability they measured which is contradicting a very recent publication by their rival Peter Hegemann's group (Vierock et al. Sci Report 2017) which detected no/minimal calcium permeability with Chrimson. Given how different they are, this result should be commented/explained.

The equation that we used for the determination of the relative calcium permeability is:

$$\frac{P_{Ca}}{P_{Na}} = \frac{e^{\frac{\Delta U_{rev} F}{RT}} [Na]_o}{4[Ca]_o}$$

The derivation of the equation is explained in the given reference (Hille, B. Ion channels of Excitable membranes. 3rd edn, 441-470). We added the sentence: "Of note, the measured cation permeabilities contradict a recent publication²⁶, but are in accordance with a previous study⁶." to our manuscript.

The neuronal traces in Supplementary Figure 9 is a bit odd in the amplitudes of action potential varies significantly (the middle trace

shows ~100mV as expected and top and bottom show ~80mV). The thresholds for action potentials (or the resting membrane potentials) also varies significant between the middle and the other two traces. Similarly in Figure 3 of the main text, the firing threshold of the parvalbumin cell in panel (a) appears to be much higher than the parvalbumin cell(s) shown in (c) (40 mV vs 10mV above resting membrane potential), is this correct or is the membrane potential very different in these cases? Can they provide evidences these are presented or measured correctly.

We present a data-set consisting of both cell-attached and whole-cell recordings of parvalbumin-positive interneurons to ensure that the conclusion of unprecedented optogenetic control of rapid action potential firing is independent of the recording approach. Although the relevance of the reviewer's query for this finding is unclear to us, we have now addressed this in the subset of experiments in which whole-cell recordings were employed. We find that the apparent action potential threshold defined as the voltage at which the first temporal derivative crosses a threshold of 40 V/s (e.g. Hu and Jonas, 2014; A supracritical density of Na⁺ channels ensures fast signaling in GABAergic interneuron axons, *Nature Neuroscience*) is indeed more hyperpolarized for optogenetically evoked action potentials (-57.00±1.85 mV) compared to action potentials during injection of current steps (-43.19±2.97 mV, p<0.001 in unpaired, two-tailed t-test). We have added the following sentence to the Methods section: 'We note that the apparent action potential threshold defined as the voltage at which the first temporal derivative crosses a threshold of 40 V/s is more hyperpolarized for optogenetically evoked action potentials (-57.00±1.85 mV) compared to action potentials during DC current injections (-43.19±2.97 mV, p<0.001, unpaired, two-tailed t-test).'

This is a well-described, general attribute of neurons: The faster the rate of rise of the membrane potential that leads to the action potential, the more hyperpolarized the action potential threshold (e.g. Bryant and Segundo, 1976; Spike initiation by transmembrane current: a white-noise analysis. *J. Physiol.*; Azouz and Gray, 1999; Cellular mechanisms contributing to response variability of cortical neurons in vivo. *J. Neurosci.*; Henze and Buzsaki, 2001; Action potential threshold of hippocampal pyramidal cells in vivo is increased by recent spiking activity, *Neuroscience*). Consistent with this being a general effect of pulsed optogenetic activation being more rapid, we have performed the same analysis on an older data-set in which action potentials were evoked in parvalbumin-positive interneurons using the Channelrhodopsin-2 version H134R (Nagel et al., 2005; Light activation of channelrhodopsin-2 in excitable cells of *Caenorhabditis elegans* triggers rapid behavioral responses, *Current Biology*) and by DC current injections and found the same difference in apparent action potential threshold (not shown). The apparent difference in action potential threshold in Supplementary Figure 9 is due to the fact that the membrane potential of this neuron was more hyperpolarized in the middle trace. However, it still clearly makes the point requested by the reviewer that irradiance was adjusted individually for precise optogenetic control. Finally, we want to point out that we have significant expertise in patch-clamp recordings in brain slices and *in vivo*. We have published 10 papers using this technique, including direct dendritic (Letzkus et al., 2006, *Journal of Neuroscience*, Davie et al., 2006, *Nature Protocols*) and axonal recordings (Kole, Letzkus, Stuart, 2007, *Neuron*) and recordings from different types of cortical interneurons (Letzkus et al., 2011, *Nature*; Wolff et al., 2014, *Nature*).

I believe the authors have misinterpreted the comments regarding membrane resistance and capacitance. As the authors are comparing the performance of the Chrimson mutants in Fig 2 e-g and Suppl table 4 with very low number of cells (n=7), it is important to show that the membrane resistance and capacitance of these 3 groups are comparable to make sure the comparison is justified (scatter plot would be preferred over bar graphs or mean +/- SD). It would be even better if they can show that expression levels are comparable but this is harder to quantify accurately so it is not as crucial. Alternatively, the authors can randomly record from higher number of cells (~15-20 in each group) to justify the analysis statistically.

Robust expression and neuronal photostimulation by comparatively low light intensities is shown in primary neuronal cultures and in the auditory system. The given membrane resistance (0.7 G Ω to 3.5 G Ω) and capacitance values (35.4 ± 12.4 pF) were determined in additional voltage clamp experiments in the presence of TTX. We do not have membrane resistance values for the neurons measured for Fig 2 e-g.

Hippocampal cells in culture are inhomogeneous, and in particular their excitability is governed, in addition to capacitance and input resistance as pointed out by the reviewer, by a number of voltage-gated channel species as well as morphology. Therefore we feel that the additional recordings suggested by the reviewer have only a very marginal potential to improve our results. The novel Channelrhodopsin mutants will not be used under conditions where all these factors are controlled but instead most impactfully *in vivo*, where none of these factors are typically known. In addition, the hippocampal data is confirmed and expanded in a more specific way in identified parvalbumin-positive, fast-spiking interneurons, which can be driven to their intrinsic action potential frequency limit using one of our Chrimson variants. Finally, the results on the inner ear show clearly the enormous increase of the frequency response of the optical Auditory Brainstem response (oABR), due to the increased speed of the new Chrimsons. To obtain these results a mass of spiral ganglion cells were illuminated, which makes the single cell statistics obsolete. In other words, we provide converging evidence from a large number of preparations indicating that our new tools are fast with an excellent expression both *in vitro* (NG108-15, primary cultures of hippocampal neurons, parvalbumin positive interneurons in acute slices) and *in vivo* (SGN's), and feel that given this strong data set an additional experimental analysis focused on hippocampal cultures will only add marginal, if any, information.

We added the sentence: "The large variability of the dependence of the spike probability on the light intensity (**Fig. 2e-g** and **Supplementary Table 4**) is likely due to expression differences as well as the variability of membrane resistance, capacitance and spiking threshold of the investigated neurons." in order to address the reviewers concerns.

The images provided in the rebuttal on the membrane expression appears to be much better (in resolution, size and clarity) than the ones provided in the insets in the main figure or the current supplemental figure. These should go into the supplemental material.

Done

Minor points on presentation

It is not totally correct that space clamp issue lead to underestimation of the current density. Space clamp issue leads to lack of clamping in distal dendrites (and axon) that can either lead to over or underestimation of the current density depending on what happens at distal processes.

Please note that we don't measure endogenous conductivities, but channelrhodopsin photocurrents. The photocurrents were measured at a membrane potential of -70 mV. Hence, a lack of clamping leads to a reduction of the driving force, which leads to a reduction of the measured photocurrent.

The use of J to represent two different values (current density and illumination power) in supplemental table 4 is very confusing.

We now use the letter I for the illumination power.

What cells are used in generating Suppl fig 6. Is this done in neurons since it is under the paragraph on neurons?

The determination of the EC50 value was done in NG108-15 cells. The missing information in the figure legend is provided now.

I should remind the authors to do a better job of editing their manuscript as stated previously. The followings are just some examples of the current version and not the complete list that needs to be addressed. In some lesser cases, they are very awkward to read, in more severe cases they are scientifically incorrect.

Can authors clarify this very long running sentence (ln 523) 'Data were acquired with a Multiclamp 700B amplifier in loose seal cell-attached (n=4) or whole-cell current-clamp mode (n=3 included into light-response data + n = 5 additionally included to the input-output curves in Fig. 3B). This is very confusing statement that needs to be rewritten in a clear manner.

We have rephrased this sentence in the Methods section as follows: 'Data were acquired with a Multiclamp700B amplifier and pClamp 10.5 software (Axon Instruments). Optogenetically evoked action potentials were recorded in parvalbumin-positive interneurons in loose-seal cell-attached (n = 4) or whole-cell current-clamp mode (n = 3). In addition, another 5 parvalbumin-positive interneurons were recorded for the input-output curves presented in Fig. 3 B.'

Ln 43 'We developed structure-based mutants leading to a unifying concept' The phasing is incorrect here. I believe they mean 'We developed a homologous structure model that allow us to develop mutants'. 'Structure-based mutants leading to a unifying concept' as a sentence makes little sense scientifically.

Done. Note the limitation of the word count for the abstract.

Ln 47 'Because red light causes low light scattering...'. The correct phasing should be 'Because red light has lower light scattering'. Light scattering is the result of the non-homogenous and non-transparent nature of tissue (so it is caused by tissue) and not because light itself 'causes' scattering.

Done.

Ln 51 'They drive spiking.....' What is meant by 'they', does this refer optical cochlear implants or Chrimson or the system the authors developed (both the implants and chrimson variant). I suspect it is the later. Please rephrase the sentence to make this understandable

Done

Ln 63 and 67 'more light'. I think it is more accurate to either refer to stronger light or more photons instead of the ambiguous 'more light'.

Done.

Ln 113 'Permeability ratios were calculated according to the Goldman-Hodgkin-Katz equation by the shift of the reversal potential due to the exchange of external sodium by calcium.' This is another long running sentence with incorrect phasing. The following suggestion can be considered (although the authors should further work on it. 'Permeability ratios were calculated using the Goldman-Hodgkin-Katz equation with the measured values of the reversal potentials after replacing external sodium by calcium.'

Done.

Ln 115 '....calcium permeability could be verified in....' maybe just use 'is also verified'.

Done.

Ln 138 'carries the K176R mutation in addition' => 'carries the additional K176R mutation'

Done.

Ln 159, Ln 169. The use of 'proved' and 'proven' should be avoided in scientific writing. Scientifically proven something would require a higher level of testing than currently presented.

Done.

Can the authors clarify the sentence 'Optical stimulation was performed by a LED coupled to the objective (coolLed)' ln 528. I assume they mean the LED was coupled to the microscope into the optical light path instead of a direct coupling of LED to the objective to excite the specimen.

We have rephrased this sentence in the Methods section as follows: 'Optical stimulation was performed through the objective by an LED (coolLED) coupled to the microscope.'

** See Nature Research's author and referees' website at www.nature.com/authors for information about policies, services and author benefits

Additional Responses to Reviewers:

Reviewers' comments:

Reviewer #1 (Remarks to the Author):

From the author's response, is it correct that 'blind' selection of cell for recording is only done in neurons but not in NG108-15 cells. The authors would need to state this in their comparison of current density of different variants (Supp table 2) that the current density measurement is subject to experimenter's selection bias as this is not corrected for expression level on the membrane and cells are selected based on observable membrane fluorescence.

The reviewer is correct that we have omitted a detailed description of how the NG cells were selected for recording, and how their current density was determined. This has now been added to the section "Electrophysiological recording on NG108-15 cells" as follows: "The current density ($J_{-60 \text{ mV}}$) was determined by dividing the stationary current in response to a 500 ms light pulse with a saturating intensity of 23 mW/mm² by the capacitance of the cell. In order to avoid an experimental bias, the NG108-15 cells for the electrophysiological recordings were chosen independent of the brightness of their EYFP fluorescence."

Can the authors check their reference on GHK equation for calculating calcium permeability. The original GHK equation does not take account of calcium due to its divalent nature which complicated the calculation. Can the authors also elaborate within their manuscript regarding the significant calcium permeability they measured which is contradicting a very recent publication by their rival Peter Hegemann's group (Vierock et al. Sci Report 2017) which detected no/minimal calcium permeability with Chrimson. Given how different they are, this result should be commented/explained.

The equation that we used for the determination of the relative calcium permeability is:

$$\frac{P_{Ca}}{P_{Na}} = \frac{e^{\frac{\Delta U_{rev} F}{RT}} [Na]_o}{4[Ca]_o}$$

The derivation of the equation is explained in the given reference (Hille, B. Ion channels of Excitable membranes. 3rd edn, 441-470). We added the sentence: "Of note, the measured cation permeabilities contradict a recent publication²⁶, but are in accordance with a previous study⁶." to our manuscript.

The neuronal traces in Supplementary Figure 9 is a bit odd in the amplitudes of action potential varies significantly (the middle trace shows ~100mV as expected and top and bottom show ~80mV). The thresholds for action potentials (or the resting membrane potentials) also varies significant between the middle and the other two traces. Similarly in Figure 3 of the main text, the firing threshold of the parvalbumin cell in panel (a) appears to be much higher than the parvalbumin cell(s) shown in (c) (40 mV vs 10mV above resting membrane potential), is this correct or is the membrane potential very different in these cases? Can they provide evidences these are presented or measured correctly.

We present a data-set consisting of both cell-attached and whole-cell recordings of parvalbumin-positive interneurons to ensure that the conclusion of unprecedented optogenetic control of rapid action potential firing is independent of the recording approach. Although the relevance of the reviewer's query for this finding is unclear to us, we have now addressed this in the subset of experiments in which whole-cell recordings were employed. We find that the apparent action potential threshold defined as the voltage at which the first temporal derivative crosses a threshold of 40 V/s (e.g. Hu and Jonas, 2014; A suprathreshold density of Na⁺ channels ensures fast signaling in GABAergic interneuron axons, *Nature Neuroscience*) is indeed more hyperpolarized for optogenetically evoked action potentials (-57.00±1.85 mV) compared to action potentials during injection of current steps (-43.19±2.97 mV, p<0.001 in unpaired, two-tailed t-test). We have added the following sentence to the Methods section: "We note that the apparent action potential threshold defined as the voltage at which the first temporal derivative crosses a threshold of 40 V/s is more hyperpolarized for optogenetically evoked action potentials (-57.00±1.85 mV) compared to action potentials during DC current injections (-43.19±2.97 mV, p<0.001, unpaired, two-tailed t-test)."

This is a well-described, general attribute of neurons: The faster the rate of rise of the membrane potential that leads to the action potential, the more hyperpolarized the action potential threshold (e.g. Bryant and Segundo, 1976; Spike initiation by transmembrane current: a white-noise analysis. *J. Physiol.*; Azouz and Gray, 1999; Cellular mechanisms contributing to response variability of cortical neurons in vivo. *J. Neurosci.*; Henze and Buzsaki, 2001; Action potential threshold of hippocampal pyramidal cells in vivo is increased by recent spiking activity, *Neuroscience*). Consistent with this being a general effect of pulsed optogenetic activation being more rapid, we have performed the same analysis on an older data-set in which action potentials were evoked in parvalbumin-positive interneurons using the Channelrhodopsin-2 version H134R (Nagel et al., 2005; Light activation of channelrhodopsin-2 in excitable cells of *Caenorhabditis elegans* triggers rapid behavioral responses, *Current Biology*) and by DC current injections and found the same difference in apparent action potential threshold (not shown). The apparent difference in action potential threshold in Supplementary Figure 9 is due to the fact that the membrane potential of this neuron was more hyperpolarized in the middle trace. However, it still clearly makes the point requested by the reviewer that irradiance was adjusted individually for precise optogenetic control. Finally, we want to point out that we have significant expertise in patch-clamp recordings in brain slices and *in vivo*. We have published 10 papers using this technique, including direct dendritic (Letzkus et al., 2006, *Journal of Neuroscience*, Davie et al., 2006,

Nature Protocols) and axonal recordings (Kole, Letzkus, Stuart, 2007, *Neuron*) and recordings from different types of cortical interneurons (Letzkus et al., 2011, *Nature*; Wolff et al., 2014, *Nature*).

I believe the authors have misinterpreted the comments regarding membrane resistance and capacitance. As the authors are comparing the performance of the chrimson mutants in Fig 2 e-g and Suppl table 4 with very low number of cells (n=7), it is important to show that the membrane resistance and capacitance of these 3 groups are comparable to make sure the comparison is justified (scatter plot would be preferred over bar graphs or mean +/- SD). It would be even better if they can show that expression levels are comparable but this is harder to quantify accurately so it is not as crucial. Alternatively, the authors can randomly record from higher number of cells (~15-20 in each group) to justify the analysis statistically.

Hippocampal cells in culture are indeed inhomogeneous, and in particular their excitability is governed, in addition to capacitance and input resistance as pointed out by the reviewer, by a number of voltage-gated channel species as well as morphology. In order to address the reviewer's concern, we performed additional experiments that strengthen the statistical basis (now 15 cells in each group) of our results and added the sentence: "The large variability of the dependence of the spike probability on the light intensity (**Fig. 2e-g** and **Supplementary Table 4**) is likely due to differences in the expression of ion channels as well as to variability of membrane resistance, capacitance and spiking threshold of the investigated neurons. "

The images provided in the rebuttal on the membrane expression appears to be much better (in resolution, size and clarity) than the ones provided in the insets in the main figure or the current supplemental figure. These should go into the supplemental material.

Done

Minor points on presentation

It is not totally correct that space clamp issue lead to underestimation of the current density. Space clamp issue leads to lack of clamping in distal dendrites (and axon) that can either lead to over or underestimation of the current density depending on what happens at distal processes.

Please note that we do not measure endogenous conductivities, but channelrhodopsin photocurrents. The photocurrents were measured at a membrane potential of -70 mV. Hence, a lack of clamping leads to a reduction of the driving force, which leads to a reduction of the measured photocurrent. In response to the reviewer's comment we have now added the following statement:

The use of J to represent two different values (current density and illumination power) in supplemental table 4 is very confusing.

We now use the letter I for the illumination power.

What cells are used in generating Suppl fig 6. Is this done in neurons

since it is under the paragraph on neurons?

The determination of the EC50 value was done in NG108-15 cells. The missing information in the figure legend is provided now.

I should remind the authors to do a better job of editing their manuscript as stated previously. The followings are just some examples of the current version and not the complete list that needs to be addressed. In some lesser cases, they are very awkward to read, in more severe cases they are scientifically incorrect.

Can authors clarify this very long running sentence (ln 523) 'Data were acquired with a Multiclamp 700B amplifier in loose seal cell-attached (n=4) or whole-cell current-clamp mode (n=3 included into light-response data + n = 5 additionally included to the input-output curves in Fig. 3B). This is very confusing statement that needs to be rewritten in a clear manner.

We have rephrased this sentence in the Methods section as follows: 'Data were acquired with a Multiclamp700B amplifier and pClamp 10.5 software (Axon Instruments). Optogenetically evoked action potentials were recorded in parvalbumin-positive interneurons in loose-seal cell-attached (n = 4) or whole-cell current-clamp mode (n = 3). In addition, another 5 parvalbumin-positive interneurons were recorded for the input-output curves presented in Fig. 3 B.'

Ln 43 'We developed structure-based mutants leading to a unifying concept' The phasing is incorrect here. I believe they mean 'We developed a homologous structure model that allow us to develop mutants'. 'Structure-based mutants leading to a unifying concept' as a sentence makes little sense scientifically.

Done. Note the limitation of the word count for the abstract.

Ln 47 'Because red light causes low light scattering...'. The correct phasing should be 'Because red light has lower light scattering'. Light scattering is the result of the non-homogenous and non-transparent nature of tissue (so it is caused by tissue) and not because light itself 'causes' scattering.

Done.

Ln 51 'They drive spiking.....' What is meant by 'they', does this refer optical cochlear implants or Chrimson or the system the authors developed (both the implants and chrimson variant). I suspect it is the later. Please rephrase the sentence to make this understandable

Done

Ln 63 and 67 'more light'. I think it is more accurate to either refer to stronger light or more photons instead of the ambiguous 'more light'.

Done.

Ln 113 'Permeability ratios were calculated according to the Goldman-Hodgkin-Katz equation by the shift of the reversal potential due to the exchange of external sodium by calcium.' This is another long running sentence with incorrect phasing. The following suggestion can be considered (although the authors should further work on it.
'Permeability ratios were calculated using the Goldman-Hodgkin-Katz

equation with the measured values of the reversal potentials after replacing external sodium by calcium.'

Done.

Ln 115 '...calcium permeability could be verified in...' maybe just use 'is also verified'.

Done.

Ln 138 'carries the K176R mutation in addition' => 'carries the additional K176R mutation'

Done.

Ln 159, Ln 169. The use of 'proved' and 'proven' should be avoided in scientific writing. Scientifically proven something would require a higher level of testing than currently presented.

Done.

Can the authors clarify the sentence 'Optical stimulation was performed by a LED coupled to the objective (coolLED)' ln 528. I assume they mean the LED was coupled to the microscope into the optical light path instead of a direct coupling of LED to the objective to excite the specimen.

We have rephrased this sentence in the Methods section as follows: 'Optical stimulation was performed through the objective by an LED (coolLED) coupled to the microscope.'

Reviewers' Comments:

Reviewer #1 (Remarks to the Author):

The authors have addressed most of the scientific points.

Minor point:

In Supplementary Table 2, the authors use this table to judge the expression levels of the variants on the membrane in results. I assume this is based on some statistical tests on specific groups given the mutants with K176R (including vf-chrimson and K176R/Y261F/S267M/Y268F) are probably not significant after adjustment for multiple comparisons? The authors may need to point out the statistical tests used, and which groups are significant from each other.

REVIEWERS' COMMENTS:

Reviewer #1 (Remarks to the Author):

The authors have addressed most of the scientific points.

Minor point:

In Supplementary Table 2, the authors use this table to judge the expression levels of the variants on the membrane in results. I assume this is based on some statistical tests on specific groups given the mutants with K176R (including vf-chrimson and K176R/Y261F/S267M/Y268F) are probably not significant after adjustment for multiple comparisons? The authors may need to point out the statistical tests used, and which groups are significant from each other.

Supplementary Table 2 shows the current densities of the Chrimson variants in NG108-15 cells, measured at saturating light intensities. Current densities, measured at saturating light intensities are a direct measure of membrane-targeted expression. For statistical analysis we used a one way ANOVA with Bonferroni post-hoc test. Chrimson mutants carrying the K176R mutation and/or the Y268F mutation have reduced current densities in NG108-15 cells. We provide the requested information in Supplementary Table 2.